# SAFETY REPRESENTATIONS FOR SAFER POLICY LEARNING

**Kaustubh Mani**[1], **Vincent Mai**[2], **Charlie Gauthier**[1], **Annie Chen**[3], **Samer Nashed**[1], **Liam Paull**[1,4]
[1]Mila, Université de Montréal, Quebec, Canada
[2] Hydro-Québec Research Center, Quebec, Canada
[3] Standford University
[4] Canada CIFAR AI Chair
`kaustubh.mani@mila.quebec`

## ABSTRACT

Reinforcement learning algorithms typically necessitate extensive exploration of the state space to find optimal policies. However, in safety-critical applications, the risks associated with such exploration can lead to catastrophic consequences. Existing safe exploration methods attempt to mitigate this by imposing constraints, which often result in overly conservative behaviours and inefficient learning. Heavy penalties for early constraint violations can trap agents in local optima, deterring exploration of risky yet high-reward regions of the state space. To address this, we introduce a method that explicitly learns state-conditioned safety representations. By augmenting the state features with these safety representations, our approach naturally encourages safer exploration without being excessively cautious, resulting in more efficient and safer policy learning in safety-critical scenarios. Empirical evaluations across diverse environments show that our method significantly improves task performance while reducing constraint violations during training, underscoring its effectiveness in balancing exploration with safety.

## 1 INTRODUCTION

Reinforcement learning (RL) has achieved notable success across various domains, from game playing (Shao et al., 2019) to robotics (Zhang & Mo, 2021; Singh et al., 2022). However, training RL agents directly in safety-critical environments remains challenging partly due to the presence of failure events that are deemed undesirable or unacceptable both during training and deployment. To manage these failures, RL algorithms typically rely on failure penalties or impose safety constraints (Xie et al., 2022; Leike et al., 2017; Achiam et al., 2017; Stooke et al., 2020). While these methods help minimize unsafe behaviours during learning, they often result in overly cautious policies, restricting the agent's ability to explore leading to sub-optimal performance. The key challenge, therefore, lies in designing RL algorithms that can effectively balance the risks of exploration with the reward of task completion.

One significant factor contributing to conservative behaviour in RL agents is "primacy bias" (Nikishin et al., 2022), where early experiences exert a lasting influence on the agent's learning trajectory. In safety-critical applications, severe penalties for constraint violations encountered early in the training process can disproportionately shape the agent's policy, leading to overly cautious decision-making and hindered exploration. As a result, agents learn safety representations that overestimate the risk of failure, discouraging further exploration. This often results in agents with a narrow view of the state space and locally optimal yet overly conservative policies that sacrifice performance for safety. While a considerable body of research has explored some notion of safety estimation for safer policy learning (Tang et al., 2019; Chow et al., 2018; Greenberg et al., 2022), most approaches focus on failure prevention by implicitly or explicitly restricting agent exploration. For example, methods such as Bharadhwaj et al. (2020) and Srinivasan et al. (2020) filter out actions with the likelihood of failure above a specific threshold. In this work, we demonstrate that by incorporating accurate safety representations into the learning process, agents can make more informed decisions, balancing exploration with safety.

Specifically, we present *Safety Representations for Policy Learning* (SRPL), a framework that augments an RL agent's state representation by integrating a state-conditioned safety representation derived from the agent's experiences during the learning process. SRPL is grounded in the understanding that risk is often unevenly distributed among states. For instance, driving in the wrong lane on a two-lane road is inherently unsafe, independent of the specific policy the agent follows. The safety representation is captured by a steps-to-cost (*S2C*) model, which for any given state estimates the distribution over the proximity to unsafe or cost-inducing states. Additionally, by learning safety representations that are state-centric utilizing data from the agent's experience (as opposed to just the current policy), we encourage the generalizability of the safety representations across tasks.

To summarize, we study three primary hypotheses addressing the key challenges outlined:

- By directly integrating safety information into the state representation, the safety and efficiency of RL agents during the learning process are significantly enhanced.

- Safety representations can be efficiently learned online using the experiences generated during RL training, resulting in improved performance without requiring prior or additional data.

- When learned from an agent's entire experience that includes a diversity of policies, safety representations can be generalized across various tasks, acting as an effective prior for learning new tasks.

The SRPL framework can be used to augment any RL algorithm and we show results for several on-policy and off-policy baselines in Sec 5. We evaluate SRPL agents on several simulated robotic tasks, including manipulation, navigation, and locomotion. Our results show that by leveraging safety information, SRPL agents are significantly more sample-efficient while being safer during learning compared to baselines. Additionally, safety information transfers well across tasks, providing a useful prior for learning new policies.

## 2 Preliminaries

**Markov decision processes (MDPs)** are defined as a tuple $\langle S, A, T, R, d, \gamma \rangle$, where $S$ is a set of states, $A$ is a set of actions, $R : S \times A \times S \to \mathbb{R}$ is the reward function indicating the immediate reward for executing action $a$ in state $s$ and resulting in state $s'$, $T : S \times A \times S \to [0, 1]$ is the forward dynamics model indicating the probability of transitioning to state $s'$ after executing action $a$ in state $s$, $d : S \to [0, 1]$ represents the probability of starting in a state $s \in S$, and $\gamma \in [0, 1)$ is the discount factor. The solution to an MDP is an optimal policy $\pi^*$ that maximizes the expected discounted cumulative reward, $J(\pi)$:

$$J(\pi) = \mathbb{E}_{s_0 \sim d(s)} \Big[ \mathbb{E}_{\tau \sim \pi} \Big[ \sum_{t=0}^{H} \gamma^t R(s_t, a_t, s_{t+1}) \Big] \Big]. \tag{1}$$

Here, $H$ is the length of the horizon.

**Constrained MDPs (CMDPs)** are defined as a tuple $\langle S, A, T, R, d, \gamma, \mathcal{C}, \beta \rangle$, where $\langle S, A, T, R, d, \gamma \rangle$ is an MDP, $\mathcal{C} : S \to \{0, 1\}$ is a cost function, and $\beta$ is the constraint threshold or a maximum, cumulative cost that is acceptable in expectation. Intuitively, we can view the constraints imposed by $\mathcal{C}$ and $\beta$ as reducing the set of possible policies from all policies $\Pi$ to those satisfying the constraints $\Pi_{\mathcal{C}} \subseteq \Pi$. The solution to a CMDP is a policy $\pi^*$ where

$$\pi^* = \text{argmax}_{\pi \in \Pi_C} J(\pi), \Pi_{\mathcal{C}} = \big\{ \pi \in \Pi : J_{\mathcal{C}}(\pi) \leq \beta \big\}.$$
$$\text{where} \quad J_{\mathcal{C}}(\pi) = \mathbb{E}_{s_0 \sim d(s)} \Big[ \mathbb{E}_{\tau \sim \pi} \Big[ \sum_{t=0}^{H} \gamma^t \mathcal{C}(s_t) \Big] \Big] \tag{2}$$

To find an optimal policy, online RL algorithms allow an agent to explore the environment while simultaneously using these trajectory rollouts to optimize the policy. In deep reinforcement learning (DRL), it is typically impossible to guarantee that an agent will never execute a policy $\pi \notin \Pi_{\mathcal{C}}$ during training. Therefore, in the absence of any prior information, the agent will likely violate

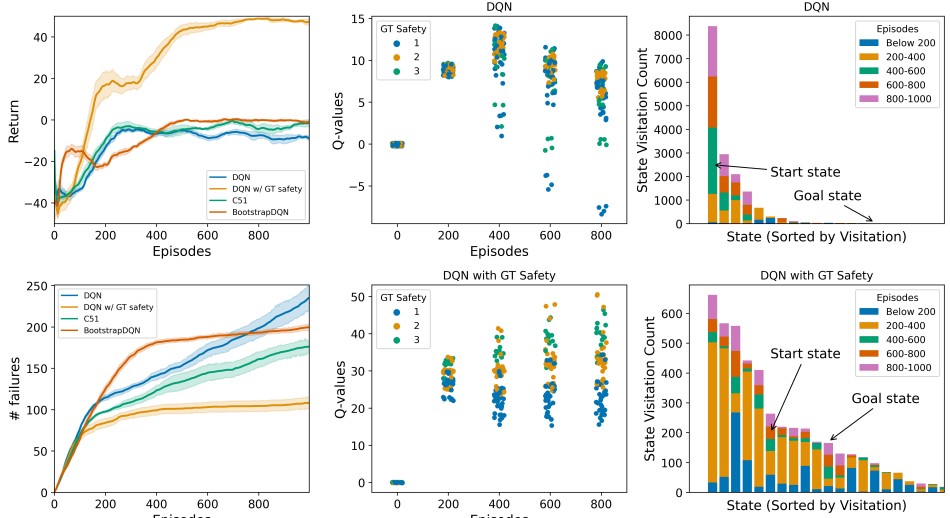

Figure 1: To motivate the benefit of learning state-conditioned safety representations in safety-critical applications, we perform experiments on the *Island Navigation* environment Leike et al. (2017). We assume access to the Manhattan distance from the nearest water cell as ground truth (GT) safety information. **(Col 1)** shows that without this information, penalties due to failure early in the learning process bias the agent toward overly conservative behaviour resulting in suboptimal policies that avoid water but fail to complete the task. **(Col 2)** compares the Q-value estimates of a DQN agent with and without GT safety information across all states over multiple episodes. Without safety information, the agent fails to distinguish between risky and less risky states, producing uniformly low Q-values across all states. This highlights the inability of RL agents to learn good safety representations using reward signals alone. **(Col 3)** examines state visitation patterns over episodes, showing that while both agents initially explore the environment, the agent without safety information quickly reduces exploration, oscillating between two states to avoid failure but failing to reach the goal state while the agent with safety information is able to explore a larger region of the state-space. More detailed discussion in Sec. 3.1

the constraints during exploration. Given this, we often care about both the cumulative constraint violations incurred during training as well as the expected constraint violations of the final policy.

In the remainder of this paper, we will refer to states that we aim to avoid, such as sink states in MDPs and cost-inducing states in CMDPs, as "unsafe" states. We've also used "distance to unsafe" and "steps to unsafe" interchangeably. We define a failure as an event where a constraint is violated, and we will use the terms "failure" and "constraint violation" interchangeably. Furthermore, following common practices in safe RL literature (Achiam et al., 2017; Stooke et al., 2020; Sootla et al., 2022), we assume that unsafe states can be identified either through the termination of an episode or via the cost signal.

## 3 SAFETY REPRESENTATIONS FOR POLICY LEARNING

In this section, we will begin by motivating the usefulness of safety information in a toy example where we assume that this information is somehow provided. Subsequently, we will formalize our choice of safety representation and describe the SRPL framework.

### 3.1 MOTIVATING EXAMPLE

To demonstrate the usefulness of state-centric safety representations, we perform experiments on *Island Navigation* (Leike et al., 2017), a grid world environment designed for evaluating safe exploration approaches. The agent's goal is to navigate the island without entering the water cells. Entering a water cell is considered unsafe and leads to a failure penalty in the form of a negative reward and episode termination. The agent is only rewarded when it visits the goal state. The input to the RL agent is the image of the entire grid. Instead of experimenting with a single environment with a fixed start and goal state (Leike et al., 2017) where the agent can simply memorize action

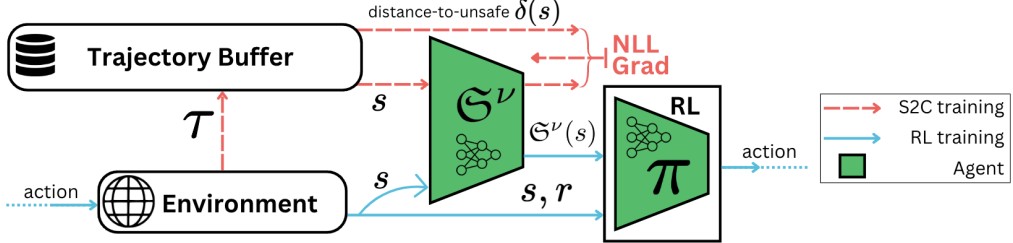

Figure 2: *SPRL Framework:* SPRL explicitly learns safety representations for states as distribution over proximity to unsafe states (cost-inducing states) through a steps-to-cost (*S2C*) model and uses this information to implicitly guide policy learning towards exploring safer regions of the state space.

sequences, we create four different versions (Fig. 11 (in the appendix)) of the *Island Navigation* environment with different start and goal positions as well as locations of the water tiles. Each episode begins with randomly selecting an environment. Thus the agent needs to learn good state-conditioned representations of safety to solve the task as well as minimize failures during learning.

In this environment, a reasonable proxy for safety associated with every cell is its Manhattan distance to the nearest water cell. To investigate how this priviliged safety information can be useful to the policy, we provide the RL agent with this distance (referred to as GT safety) by adding it to the state representation. We train DQN(Mnih et al., 2015) agents with and without this safety information, along with variants of DQN that model aleatoric (c51 Bellemare et al. (2017)) and epistemic (BootstrapDQN Osband et al. (2016)) uncertainty.

Fig. 1 (col 2) shows the evolution of the Q-value distributions for all state-action pairs for a particular instance of the environment during training, for DQN agents with and without safety information. In the presence of safety information, the DQN agent is able to efficiently learn a correlation between safety information and reward, outputting low Q-values for risky states (GT safety = 1) and high Q-values for safe states (GT safety = 3), as a result efficiently exploring the state-space and ultimately converging to an optimal policy. On the other hand, the DQN agent without this information fails to learn accurate internal representations of safety and instead overestimates risk for all states resulting in uniformly low Q-values (Fig. 1 (col 2) (row 1)), further discouraging exploration and as a result converging to a sub-optimal policy (Fig. 1 (col 1) (row 1)). Fig. 1 (col 3) (row 1) shows that the policy learned by the DQN agent in the absence of safety information ensures safety of the agent by oscillating between two or three states near the start state[1] but fails to consistently reach the goal state resulting in low average return (Fig. 1 (col 1)).

This example demonstrates that having access to additional information about the safety of a state can be useful for RL agents to overcome the bias created by negative experiences early in learning that result in an overestimation of risk leading to low Q-values and, as a result, yield conservative policies with sub-optimal performance. However, typically this information is not provided to a learning agent and must be inferred from the agent's observation of the environment. We propose explicitly learning safety representations using agent experience during learning.

## 3.2 STATE-CENTRIC SAFETY REPRESENTATIONS

We propose to learn state-conditioned safety representations as an inductive bias to enable safety-informed agents by leveraging the agent's prior experience. This raises two key questions: (1) What constitutes an ideal state-centric representation of safety? (2) How should we train such a representation?

An effective safety representation must capture the immediate likelihood of failure in the current state as well as reflect potential risks in future states. This representation should incorporate both the risks associated with exploration from a given state, the uncertainties in the environment's dynamics, and the ambiguities in policy choices for action selection following the current state. A simple scalar safety representation (Bharadhwaj et al., 2020; Srinivasan et al., 2020), lacks the expressiveness necessary to capture these complexities. Therefore, we propose modelling safety as a probability

---

[1]Episodes terminate when the agent reaches the goal state or enters a water cell. However, oscillating between two states causes episodes to truncate at the max-steps limit of 100, resulting in high state-visitation counts

distribution over distances to cost-inducing states. To avoid learning representations that overfit data from a narrow region of the state space induced by a particular policy, we propose to learn safety representations over the entirety of the agent's experience instead of policy-specific rollouts.

Formally, we model safety as a function $\mathfrak{S} : \mathcal{S} \rightarrow \Delta^{H_s}$, where $\mathcal{S}$ is the state space and $\Delta^{H_s}$ represents a probability simplex over the safety horizon $H_s$[2]. Given the experience of an agent, the model $\mathfrak{S}_t(s)$ aims to capture the conditional probability of entering an unsafe state in exactly $t \in \{1, 2, \ldots, H_s\}$ steps given that the agent is in state $s$ (Fig. 3). Here, $\mathfrak{S}_{H_s}(s)$ represents the probability that the agent remained safe throughout the safety horizon $H_s$ without encountering an unsafe state based on the agent's past experience. This model can be learned from the set of trajectories which represent the experience of the agent, as we will describe next.

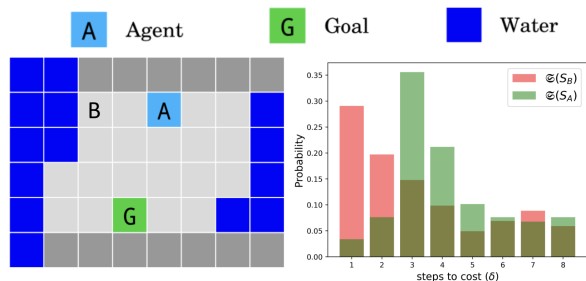

Figure 3: *Safety Representation:* We demonstrate the *S2C* model's output on two different states ($A$ & $B$). A being farther from the water cells (indicated in blue) has its peak at 3 (distance from the unsafe set) while $B$ is a more risky state and has its peak at 1.

### 3.3 SRPL FRAMEWORK

To learn the state-centric safety representation , we train a neural network, referred to as the "steps-to-cost" or the *S2C* model, which takes the state as input and outputs the corresponding safety representation. This representation is modelled as a discrete distribution, implemented as the softmax output of a neural network $\mathfrak{S}^\nu$ parameterized by $\nu$.

The safety representation is learned alongside the RL policy by constructing a separate replay buffer $\mathcal{D}_{S2C}$ in the case of on-policy algorithms, which contains trajectories $\tau = (s_0, \delta_\tau(s_0)), \ldots, (s_n, \delta_\tau(s_n))$ from policy rollouts. In the case of off-policy algorithms like CSC (Bharadhwaj et al., 2020) and CVPO Liu et al. (2022), the off-policy buffer is used to store the "steps-to-cost" information corresponding to states for each trajectory. At the end of each episode, every state $s$ in the trajectory $\tau$ is labelled with its corresponding "steps-to-cost" value $\delta_\tau(s)$, which is the number of actions taken before encountering an unsafe state. If no unsafe state is encountered during the episode, the distance to unsafe for all states in the trajectory is set to the safety horizon length ($H_s$) to indicate safety of the state within the safety horizon. In this way, for every trajectory $\tau \in \mathcal{D}_{S2C}$ and for every state $s \in \tau$, we have a label $\delta_\tau(s)$ which we can use to train our *S2C* model by minimizing the following negative log-likelihood loss[3]:

$$\mathcal{L}_{S2C}((s, \tau); \nu) = -\sum_{t=1}^{H_s} \mathbb{I}[\delta_\tau(s) = t] \log(\mathfrak{S}_t^\nu(s)) \tag{3}$$

where $\mathbb{I}[\delta(s) = t]$ is an indicator function that takes value 1 if $\delta_\tau(s) = t$ and 0 otherwise.

In the SRPL framework, (Fig. 2), the output of the *S2C* model $\mathfrak{S}^\nu(s)$ is incorporated into the agent's state by augmenting the original state with the learned safety representation. The augmented state is defined as $s' = \{s, \mathfrak{S}^\nu(s)\}$. For high-dimensional observations, such as raw images, the safety representation is concatenated with the encoded feature representation of the observation, such that the augmented observation becomes $o' = \{\mathcal{F}(o), \mathfrak{S}^\nu(o)\}$, where $\mathcal{F}$ is the feature encoder.

**Implementation details:** We model the safety distribution over a fixed safety horizon $H_s << H$, relying on the assumption that information about near-term safety is more important and an agent can safely navigate the state space with this information. Instead of modelling the distribution over all time steps between $[1, H_s]$, we split this range into bins to further reduce the dimensionality of the safety representation. An ablation over the choices of bin size and safety horizon $H_s$ is included in the Appendix A.6.1. For on-policy algorithms, a separate off-policy replay buffer is maintained which

---

[2]$H_s << H$, where H is the MDP time horizon.
[3]For categorical distributions, NLL loss is equivalent to Cross-Entropy loss.

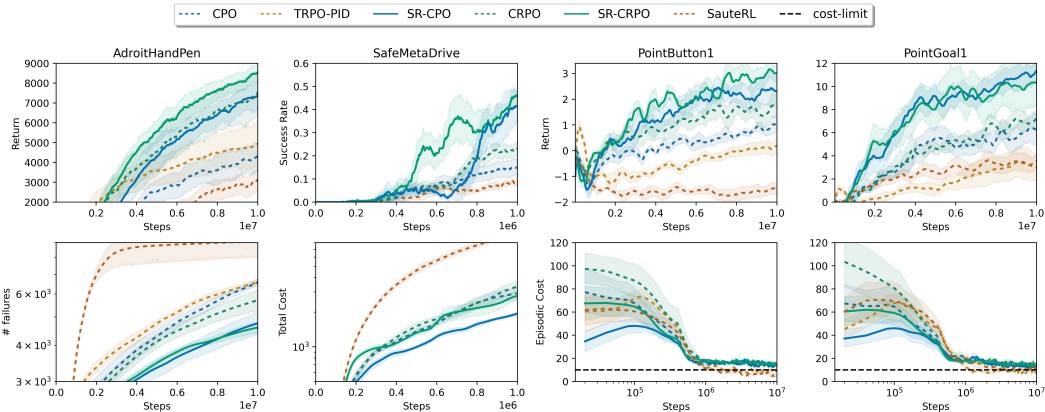

Figure 4: Performance of SRPL agents (denoted SR-*) on four different tasks. For these experiments, both the *S2C* model and the policy have been randomly initialized so no prior information has been provided to the agent. SRPL agents consistently outperform their baseline counterparts on both safety during learning as well as sample efficiency. Results were obtained by averaging the training runs across five seeds. The input to the RL agent is state-based in the form of joint states or LiDAR points.

stores policy rollouts along with the distance to unsafe values. To preserve only relevant experiences about policies similar to the agent's current policy, the replay buffer throws away samples from older policies. More thorough implementation details are provided in Appendix A.2

# 4 EXPERIMENTS

## 4.1 EXPERIMENTAL SETUP

We perform our experiments on four tasks in three distinct environments. First is a manipulation task *AdroitHandPen* (Rajeswaran et al., 2017) where a 24-degree of freedom Shadow Hand agent needs to learn to manipulate a pen from a start orientation to a randomly sampled goal orientation. Dropping the pen from the hand is considered a failure or constraint violation. Next, we have an autonomous driving environment *SafeMetaDrive* (Li et al., 2022), where an RL agent is learning to drive on the road while avoiding traffic, each collision incurs a cost and the cost threshold is set to 1 ($\beta = 1$). Finally, we evaluate our method on the Safety Gym (Ray et al., 2019) environment on tasks *PointGoal1* and *PointButton1*. For the *PointGoal1* task we have a point agent that is tasked with reaching a random goal position from a random start position, the environment consists of regions that are unsafe and accumulate cost. For the *PointButton1* task, the agent needs to press a sequence of buttons in the correct order, pressing the wrong button incurs a cost. In addition, there are static hazards and dynamic objects that the agent needs to avoid. The goal in these environments is to do the task while incurring costs lower than a threshold ($\beta = 10$). Additionally, we also show results on Mujoco locomotion environments (Ant, Hopper and Walker2d) in the Appendix. The goal in these tasks is for the robot to move as fast as possible while avoiding falling on the ground which is considered a failure or a constraint violation.

Further, we evaluate the transferability of learned safety representations from one task to another. For this, we use *PointButton1* as the source task and *PointGoal1* as the target task. We augment the state representation of the Safety Gym environments to ensure the same state dimensionality for *PointGoal1* and *PointButton1* (More details in Appendix A.4). The *S2C* model is trained on the source task and frozen for the target task and we study the performance of the *S2C* model at transferring information in terms of safety and efficiency on the target task. Additionally, we show that the *S2C* model provides a good initialization for fine-tuning representations on the target task in the case of transferring both policy and safety representations.

## 4.2 BASELINES & COMPARISON

The SRPL framework can be integrated with any RL algorithm that accounts for risk sensitivity or operates in safety-critical environments. We evaluate its effect on feature learning by comparing several baseline algorithms with their SRPL-augmented versions. We compare against both off-policy and on-policy safe RL algorithms. On-policy baselines include Constrained Policy Optimization (CPO) (Achiam et al., 2017), a second-order method for enforcing constraints, TRPO-PID (Stooke et al., 2020) a Lagrangian-based method that uses a PID controller for stable learning, SauteRL (Sootla

| Methods | AdroitHandPen | | SafeMetaDrive | | PointGoal1 | | PointButton1 | |
|---|---|---|---|---|---|---|---|---|
| | Return (↑) | #failures (↓) | Success Rate (↑) | Total Cost (↓) | Return (↑) | Cost-Rate ($*1e^2$) (↓) | Return (↑) | Cost-Rate ($*1e^2$) (↓) |
| CPO | $4154 \pm 1798$ | $6667 \pm 363$ | $0.088 \pm 0.1$ | $2715 \pm 366$ | $6.25 \pm 3.28$ | $1.61 \pm 0.12$ | $0.91 \pm 0.70$ | $1.64 \pm 0.06$ |
| TRPO-PID | $4809 \pm 2641$ | $6629 \pm 445$ | $0.076 \pm 0.17$ | $2471 \pm 1326$ | $2.96 \pm 2.61$ | $1.09 \pm 0.02$ | $0.15 \pm 0.73$ | $1.13 \pm 0.06$ |
| SauteRL | $3831 \pm 1783$ | $9100 \pm 2557$ | $0.08 \pm 0.09$ | $10170 \pm 1196$ | $3.63 \pm 1.73$ | $1.08 \pm 0.02$ | $-1.45 \pm 0.56$ | $1.09 \pm 0.02$ |
| CRPO | $7893 \pm 2576$ | $5752 \pm 1146$ | $0.22 \pm 0.24$ | $2966 \pm 732$ | $7.49 \pm 1.56$ | $1.58 \pm 0.13$ | $1.47 \pm 0.68$ | $1.58 \pm 0.05$ |
| SR-CPO | $7176 \pm 1392$ | $4797 \pm 931$ | $0.47 \pm 0.34$ | $\mathbf{1997 \pm 151}$ | $\mathbf{11.63 \pm 2.28}$ | $1.49 \pm 0.03$ | $2.21 \pm 1.16$ | $1.53 \pm 0.05$ |
| SR-TRPO-PID | $5963 \pm 1384$ | $5311 \pm 741$ | $0.12 \pm 0.29$ | $2001 \pm 1014$ | $7.31 \pm 2.85$ | $1.07 \pm 0.02$ | $1.38 \pm 1.19$ | $1.08 \pm 0.02$ |
| SR-SauteRL | $4094 \pm 1007$ | $6316 \pm 1634$ | $0.15 \pm 0.12$ | $8682 \pm 814$ | $4.46 \pm 1.25$ | $\mathbf{1.05 \pm 0.01}$ | $-1.01 \pm 0.61$ | $\mathbf{1.04 \pm 0.01}$ |
| SR-CRPO | $\mathbf{8800 \pm 985}$ | $\mathbf{4626 \pm 447}$ | $\mathbf{0.53 \pm 0.18}$ | $2889 \pm 549$ | $10.49 \pm 4.64$ | $1.51 \pm 0.02$ | $\mathbf{3.12 \pm 0.99}$ | $1.53 \pm 0.02$ |

Table 1: Performance of the RL policies at the end of the training. SRPL versions of the baseline algorithms (denoted by SR-*) consistently reduce the number of failures or cost-rate while significantly improving return. Training details for SRPL and baselines are provided in Appendix A.2. The input to the RL agent is state-based in the form of joint states or LiDAR points.

et al., 2022) a state-augmentation method that stores the remaining constraint budget as part of the state information, CRPO (Xu et al., 2021) a primal approach which updates the policy by alternating between reward maximization and constraint satisfaction. Off-policy baselines include Conservative Safety Critics (CSC) (Bharadhwaj et al., 2020) which learns a safety critic using a safe Bellman operator and Constraint Variational Policy Optimization (CVPO) (Liu et al., 2022) which formulates the constrained MDP problem as an Expectation Maximization (EM) problem.

**Evaluation Metrics.** For *AdroitHandPen* environment, we evaluate SRPL and baselines on episodic return (sample efficiency) and total failures/costs incurred during the training of the RL agent (safety). For the *SafeMetaDrive* environment, we evaluate the algorithms on task performance and safety in terms of the success rate and total cost, respectively. For the Safety Gym environments, we evaluate the algorithms on the task performance and safety in terms of episodic return and episodic cost, respectively. Additionally, we measure the task performance and safety at the end of training for the Safety Gym environments in terms of average return and cost rate respectively. Cost-rate is measured by dividing the total cost incurred during training by the number of actions/steps taken in the environment.

## 5 RESULTS

### 5.1 LEARNING SAFETY REPRESENTATIONS ALONGSIDE POLICY

We study the performance of SRPL agents when jointly training the *S2C* model and policy (Fig. 4). In this setting, the *S2C* model does not contain prior information and thus must learn state-conditioned safety representations using rollouts generated by the agent during training.

Table 1 shows the results for on-policy safe RL algorithms along with their SRPL counterparts. Learning safety representations alongside the policy greatly enhances the sample efficiency of the baseline algorithms on all tasks, while enabling faster constraint satisfaction or fewer failures or costs during training. From Fig. 4, we can see that SauteRL (Sootla et al., 2022) performs the best in terms of constraint satisfaction in the case of Safety Gym (Ray et al., 2019) environments where the constraint threshold is greater than 1 ($\beta > 1$) and fails to do the same for tasks like *AdroitHandPen* and *SafeMetaDrive*. We hypothesize that this limitation arises from how SauteRL encodes the remaining safety budget into the state. In these environments, the safety budget remains 0 until a constraint violation occurs, rendering it ineffective for accurately representing safety throughout the task.

Fig. 5 shows the results for off-policy baselines and their SRPL counterparts on Safety Gym environments *PointGoal1* and *PointButton1* over 2M timesteps[4]. By filtering out actions that might cause a constraint violation through the use of a safety critic, CSC (Bharadhwaj et al., 2020) enables better constraint satisfaction but results in conservative or suboptimal task performance. On the other hand, CVPO (Liu et al., 2022) is significantly more sample efficient than CSC as well as the on-policy counterparts. SRPL significantly improves the sample efficiency of both CVPO and CSC baselines reaching similar or higher performance in comparison to SR-CVPO and SR-CPO in just $2M$ steps.

### 5.2 RISK-REWARD TRADEOFF

In CMDPs, safety and task performance can be treated as separate objectives, framing the problem as a multi-objective optimization task. Depending on the safety-critical nature of a system, different levels

---

[4]Off-policy methods are generally more sample efficient and require fewer training samples

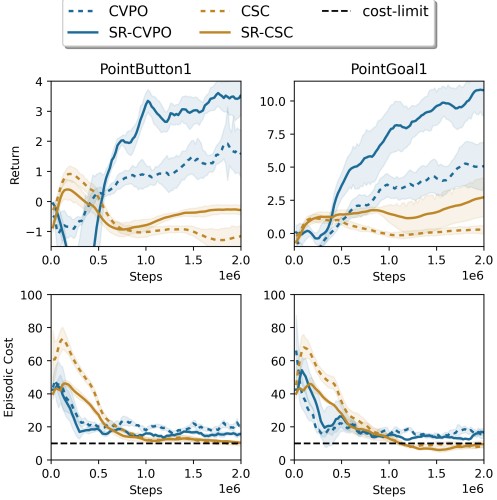

Figure 5: Off-policy results for CSC and CVPO and their SRPL counterparts on Safety Gym environments over $2M$ timesteps. While CSC has better constraint satisfaction it also leads to suboptimal performance, CVPO has better sample efficiency which is further improved by SRPL.

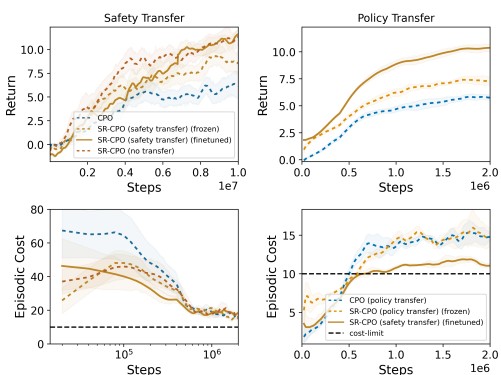

Figure 6: *Transferring the Safety Representations:* Transferring the policy as well as learned safety representations from *PointButton1* to *PointGoal1*. Zero-shot transfer of the (frozen) safety representation leads to significant improvement in terms of sample efficiency over vanilla CPO with or without policy transfer. Fine-tuning the safety representations on *PointGoal1* further improves the performance while leading to better constraint satisfaction for policy transfer.

of tolerance for constraint violations during learning can be set, effectively prioritizing one objective over the other. Emphasizing safety during the learning process tends to discourage exploration, which can subsequently reduce task performance, a phenomenon often referred to as the risk-reward or safety-performance tradeoff. To investigate the effect that learning safety representations has on this tradeoff, we conduct experiments comparing the risk-reward tradeoff capabilities of SPRL with baseline algorithms.

In algorithms that approach the CMDP problem through a Lagrangian formulation (CSC in Fig. 7), prioritizing safety corresponds to increasing the value of the initial Lagrange multiplier. A higher Lagrange multiplier value reduces exploration and enhances safety during learning. In algorithms like CPO (Achiam et al., 2017) that enforce exact constraint satisfaction at each step, we adjust the constraint threshold to modulate this tradeoff. Our experiments are conducted in two distinct environments: *AdroitHandPen* and *Ant*. In Fig. 7, each point represents the policy's total cost incurred during learning (x-axis) and its final performance (y-axis), measured as the average return, for various safety requirement settings (either Lagrange multiplier or constraint threshold). The ellipses represent the variance across both the x and y directions. As we increase the priority of safety while learning, the number of failures reduces along with the task performance for all RL agents.

Figure 7 clearly illustrates that SPRL improves baseline algorithms' ability to balance task performance and safety during the learning process. This indicates that the SPRL framework facilitates safer learning for a given level of task performance or enhances task performance for a desired level of safety. Additionally, as expected, the figure shows that safety information becomes increasingly valuable as the safety-criticality of the objective rises.

### 5.3 SRPL AS AN EFFECTIVE PRIOR

As described in Sec. 4.1, to study the generalizability of the learned safety representations across tasks, we use *PointButton1* as the source task and *PointGoal1* as the target task. We present two sets of results 1) where we transfer the safety representations but not the policy (Fig. 6(left)), training policy from scratch on the target task) and 2) where we transfer both the policy and the safety representations to the target task. Additionally, for both these cases, we study the effect of freezing the *S2C* model (i.e. safety representations) as well as finetuning the safety representations on the target task.

From Fig. 6 (left), we see that transferring safety representations without finetuning them on the target task (SR-CPO (safety transfer) (frozen)) improves the sample efficiency as well as enables

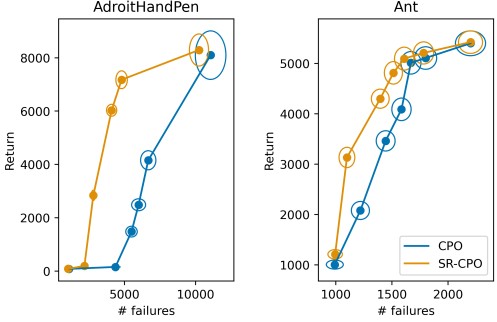

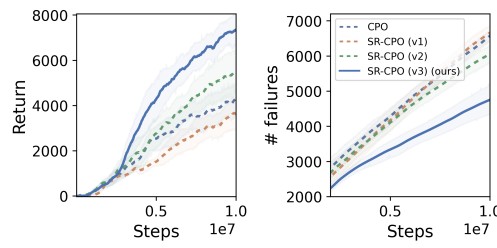

Figure 8: *Alternative Safety Representations:* An analysis of the effect of different choices in modelling safety representations on overall performance in terms of safety and efficiency of the algorithm.

Figure 7: *Risk-Reward Tradeoff:* SR-CPO is able to better tradeoff risk and reward in comparison to vanilla CPO thanks to learned safety representations.

faster constraint satisfaction on the target task. However, training the safety representations directly on the target task from scratch (SR-CPO (no transfer)) leads to better performance at convergence, due to the fact that the source and the target environments are not identical in their distribution of cost-inducing states. We further see that fine-tuning the safety representations (SR-CPO (safety transfer) (finetuned)) achieves similar performance at convergence and faster constraint satisfaction in comparison to learning representations from scratch. Fig. 6 (right) shows the results when transferring the policy from source to target task. We see that CPO with policy transfer is able to be more sample efficient compared to CPO trained from scratch. Transferring both the learned safety representations and the policy keeping the safety representations frozen (SR-CPO (policy transfer) (frozen)) leads to significant improvement in sample efficiency over CPO (policy transfer). Additionally, finetuning the safety representations on the target task (SR-CPO (policy transfer) (finetuned)) leads to even more sample-efficient agents as well as better constraint satisfaction.

# 6 ABLATIONS & ANALYSIS

## 6.1 ALTERNATIVE SAFETY REPRESENTATIONS

The choice of modelling the safety representation depends on the properties we desire our safety representations to encode. For the safety representation to be state-centric we need it to be trained in a policy-agnostic way (i.e. unlike the value function the safety representation encodes the likelihood of failure from data collected from a diverse set of policies experienced during learning). In this section, we study some of the alternate choices of safety representations:

(v1) modelling safety representation as the expected likelihood of entering an unsafe/cost-inducing state for the current policy (Bharadhwaj et al., 2020),

(v2) policy-dependent safety representation as a distribution over proximity to unsafe states trained using on-policy rollouts,

(v3) learning the safety representation, as proposed here, from the past experience of the agent, across a diverse set of policies.

As shown in Fig. 8, SRPL outperforms all other safety representation models. The safety representation learned in (v1) does not improve the performance of the RL agent, as this information is already captured by the cost-critic in CPO (Achiam et al., 2017). Although the policy-dependent safety representation (v2) enhances the base algorithm's performance, it does not surpass SRPL. We attribute this to two key factors: (1) SRPL utilizes data from previous policies, allowing the safety representation to encode information about a broader region of the state space, while the policy-dependent safety representation is confined to on-policy rollouts, a common limitation in on-policy versus off-policy reinforcement learning (Haarnoja et al., 2018); (2) by learning state-centric features, SRPL promotes more stable dynamics during RL training, in contrast to the policy-dependent representations, which fluctuates with policy updates.

## 6.2 Effect of safety representations for high-dimensional observations

To investigate the effects of learning low-dimensional state-centric safety representations from high-dimensional observations, we conducted experiments in the Safety-Gym *PointGoal1* environment Ray et al. (2019). We examined how different sensor modalities impact task performance and safety during learning. As shown in Table 2, increasing the dimensionality of sensor observations, from LiDAR to depth to RGB, leads to a decline in both safety and task performance. This degradation is likely due to the increased complexity involved in learning effective representations from higher-dimensional data. Additionally, Table 2 emphasizes that as the dimensionality of observations rises, learning safety representations as an inductive bias becomes increasingly critical for ensuring safe and efficient policy learning.

## 7 Related Work

*Representation Learning for RL:* RL algorithms often need to learn effective policies based on observations of the environment, rather than having direct access to the true state. These

| Sensor | Return ($\uparrow$) | | | Cost-Rate ($*1e^2$) ($\downarrow$) | | |
|---|---|---|---|---|---|---|
| Modality | CPO | SR-CPO | change (%) | CPO | SR-CPO | change (%) |
| LiDAR | $6.45 \pm 2.91$ | $11.63 \pm 2.28$ | $+77.69$ | $1.61 \pm 0.12$ | $1.49 \pm 0.03$ | $-19.67$ |
| Depth | $3.78 \pm 2.41$ | $9.44 \pm 3.86$ | $+149.73$ | $1.91 \pm 0.48$ | $1.66 \pm 0.1$ | $-27.47$ |
| RGB | $2.54 \pm 1.838$ | $8.134 \pm 4.46$ | $+219.98$ | $2.0 \pm 0.334$ | $1.7 \pm 0.374$ | $-30.01$ |

Table 2: Safety representation learning improves agent performance (return) and reduces constraint violations (cost-rate) in higher-dimensional observation spaces, where representation learning is typically more challenging.

observations can come from sensors like RGB cameras, LiDAR, or depth sensors. Learning representations that capture the essential aspects of the environment or task can significantly enhance efficiency and performance. To achieve this, various methods employ auxiliary rewards or alternative training signals (Sutton et al., 2011; Jaderberg et al., 2016; Riedmiller et al., 2018; Lin et al., 2019; Bhatia et al., 2023). One effective approach is learning to predict future latent states, which has proven valuable in both model-free (Munk et al., 2016; Schwarzer et al., 2020; Ota et al., 2020) and model-based (Watter et al., 2015; Ha & Schmidhuber, 2018) settings. In this paper, we've focused on learning representations for state-conditioned safety that can enable more informed decision-making in safety-critical applications.

*Safe Exploration:* Safe exploration (Achiam et al., 2017; Liu et al., 2022; Sootla et al., 2022; Stooke et al., 2020; Jiang et al., 2023; Gu et al., 2024) approaches need to contend with both the aleatoric uncertainty of the environment and the epistemic uncertainty associated with the exploration of unseen parts of the state-space. These methods commonly achieve this by restricting exploration to parts of the state space with low epistemic uncertainty. Bayesian model-based methods (Berkenkamp, 2019), represent uncertainty within the model via Gaussian processes, favouring exploration in states with low uncertainty. (Huang et al., 2023) incorporate lagrangian-methods into world models. Fisac et al. (2019), Srinivasan et al. (2020), Kang et al. (2022) and Bharadhwaj et al. (2020) use a safe Bellman operator (called the safety critic) to evaluate the risk of failure from a given state taking a particular action and use it to restrict exploration by filtering out actions with high risk of failure or formulating constraints according to the safety critic. (Sootla et al., 2022; Jiang et al., 2023) use accumulated cost as a proxy for risk associated with a state and use it to augment the state space.

## 8 Discussion & Conclusion

While SRPL offers a valuable approach for improving safety representations in RL agents, it comes with certain limitations, which, though typical of many RL methods, are still worth noting. First, SRPL has difficulty capturing long-horizon causal mechanisms related to safety. For example, a state at the beginning of a long single-way track leading to an unsafe state is actually very risky for the agent but would be represented by the safety representation as low risk or safe, as the number of actions separating the current and final, unsafe state is large. While challenging, such examples are not typical for most embodied or otherwise high-risk agents. Second, our chosen definition of safety has practical limits. We do not consider degrees of safety, and in general defining safety, harm, danger, or any other bad outcome typically involves substantial nuance in real-world settings which most safe RL methods, including SRPL, struggle to capture completely.

In summary, this paper addresses the problem of reinforcement learning (RL) agents becoming overly conservative as a result of penalties due to safety violations early in training in safety-critical environments, leading to suboptimal policies. To tackle this, we propose a framework that learns state-specific safety representations from the agent's experiences. By integrating this safety information into the state representation, our approach enables more informed and balanced decision-making.

ACKNOWLEDGEMENTS

The authors thank Yann Pequignot and Adriana Knatchbull-Hugessen for insightful discussions and useful suggestions on the early draft. This work was supported by the DEEL Project funded by the Natural Sciences and Engineering Research Council of Canada (NSERC) and the Consortium for Research and Innovation in Aerospace in Québec (CRIAQ).

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

# A APPENDIX

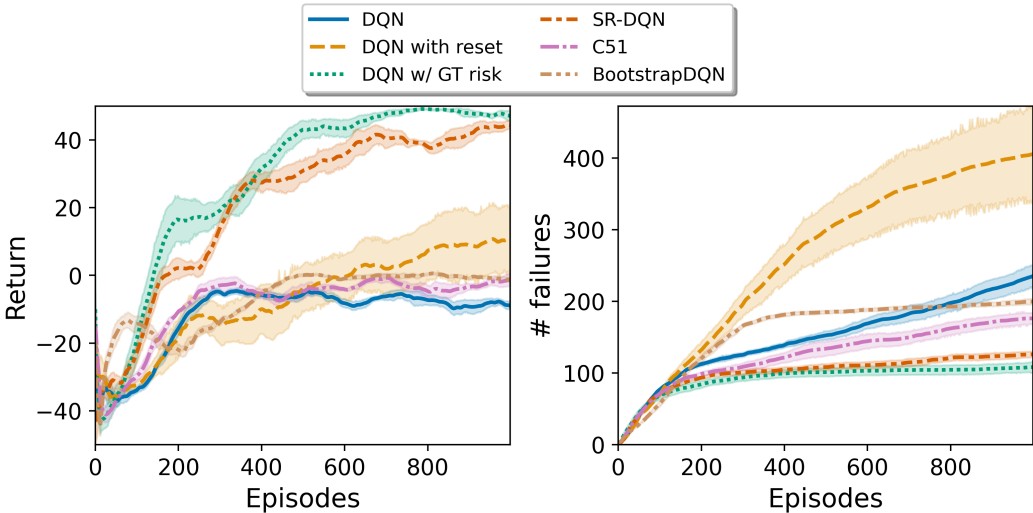

Figure 9: *DQN with resets*: Resetting the DQN agent to overcome conservatism results in significantly more failures during training.

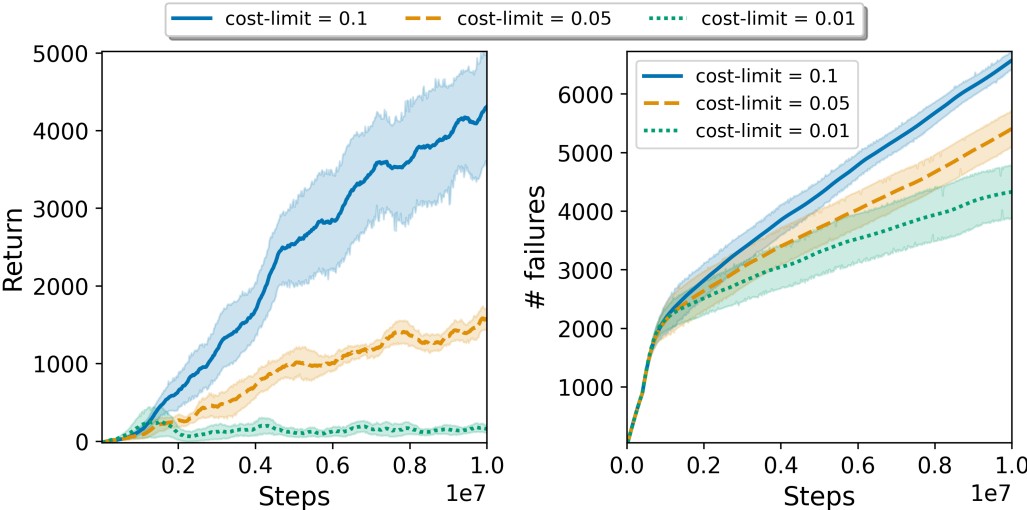

Figure 10: *Conservatism in CPO*: As the cost-limit is reduced CPO agent struggles to overcome the bias induced by initial experiences where the agent encountered heavy penalties due to constraint violation, resulting in overly conservative behaviours.

## A.1 PRIMACY BIAS IN SAFE RL

RL agents overfitting on early experience is a well-studied problem. Nikishin et al. (2022) first showed that outcomes of early experience can have long-lasting effects on subsequent learning of RL agents. Machado et al. (2018) identified poor data diversity caused by limited exploration as the primary reason for overfitting. In safety-critical systems, penalties imposed on RL agents due to constraint violations early in the training can further disincentivize exploration limiting the data diversity further. As a result RL agent tends to learn from samples from a narrow region of the state space resulting in conservative behaviors and suboptimal performance. Nikishin et al. (2022) proposed to reset the agent to encourage exploration and avoid primacy bias but for safety-critical applications resetting the RL agent can lead to catastrophic consequences. As shown in Fig. 9, DQN with resets lead to significantly more failures than vanilla DQN, while not resulting in significant

improvement in performance. In this paper, we've proposed to deal with this problem by learning state-centric representations of safety that force the agent to learn safety representations from limited data and overcome conservative behaviour.

To illustrate the extent of overfitting on negative experiences experienced by the agent early in training, we study the impact of penalties caused by constraint violation on the behaviour of the agent. We experiment on *AdroitHandPen* with CPO (Achiam et al., 2017) for different cost thresholds. A lower cost limit in the case of CPO is essentially equivalent to penalizing failures during learning more and more. From Fig. 10 We see that as we reduce the cost limit the number of failures reduces but also the resultant policy becomes more and more conservative, ultimately for cost limit = 0.01, the agent fails to learn anything about the task and settles for a local optima in which it doesn't move.

### A.2 IMPLEMENTATION DETAILS

In this section, we aim to clarify some of the design choices and implementation details used for training SRPL agents for on-policy and off-policy baselines. Our implementations were based on top of FSRL Liu et al. (2024) and Omnisafe Ji et al. (2024) codebases.

#### A.2.1 LEARNING THE SAFETY REPRESENTATION

The safety representation aims to characterize a state with respect to its proximity to the unsafe states and models a distribution over steps-to-cost based on the agent's past experience. Empirically, in the case of discrete state-spaces like *Island Navigation*, for a given state safety representation this is equivalent to maintaining the normalized frequencies over proximity to unsafe states based on the agent's past experience or policy rollouts in the past.

Here we face a dilemma about how to learn such a representation. Should we use on-policy rollouts or should we use off-policy data? There is an inherent tradeoff in this choice: if we train the safety representation using on-policy rollouts to ensure coverage we would need to collect lots of samples which will make the algorithm highly sample inefficient and also unsafe. This is particularly infeasible in the case of off-policy baselines like CVPO (Liu et al., 2022) and CSC (Bharadhwaj et al., 2020), where the policy is updated at every step. On the other hand, if we learn the safety representations completely using all of the agent's past experience, at some point the representation will become irrelevant to the current policy because it is storing information about policies that might be too different from the current policy. To manage this tradeoff, we learn the safety representation using off-policy data from the agent's past but recent experience, we maintain a separate replay buffer in case of on-policy algorithms or reuse the existing replay buffer of off-policy algorithms to store steps-to-cost $\delta_\tau(s)$ for a state encountered in a particular trajectory $\tau$. We keep a fixed replay buffer size and experiences that are generated by the older policies are discarded when the replay buffer gets full in a First In First Out (FIFO) manner. This allows us to ensure that the safety representation is learned with enough data to ensure coverage over the state space while also keeping the representation relevant for the current policy.

### A.3 TRAINING THE *S2C* MODEL

Because the output of the *S2C* model is given to condition policy learning, there are some considerations while training the *S2C* model to stabilize the training of the SRPL agent. In the case of on-policy baselines, since the policy is frozen while collecting on-policy rollouts, the dynamics of learning the safety representation alongside the policy is more stable and we update the safety representation (i.e., the *S2C*) model at a higher frequency than the policy. On the other hand, doing this in case of off-policy algorithms like CSC Bharadhwaj et al. (2020) and CVPO Liu et al. (2022) really destabilizes training because the policy is being updated at a very high frequency making the dynamics really complex. To solve this, we train the safety representations at a significantly lower frequency than the policy. This ensures that the *S2C* model is frozen while the policy is being updated. The idea is similar to the use of target networks proposed in Mnih et al. (2015) to stabilize training.

#### A.3.1 PRACTICAL CONSIDERATIONS

There are some practical considerations in order to show results on environments like *AdroitHandPen* and Mujoco (*Ant, Hopper, Walker2d*). Training these tasks with a cost-limit of 0 leads to overly

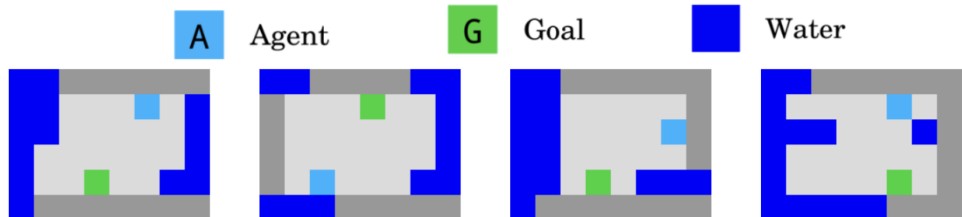

Figure 11: *Island Navigation:* Instead of experimenting with a single environment with fixed start and goal state where the agent can simply memorize action sequences. We create four different versions of the Island Navigation environment with different start and goal positions as well as locations of the water tiles.

conservative policies that learn nothing about the task as demonstrated in Fig. 7 and Fig. 10, for a cost-limit of 0, the *# failures* is very low but also the performance degrades. To address this issue we went with the cost limit of 0.1 for our experiments for both SRPL and vanilla versions of the baseline algorithms.

### A.3.2 HYPERPARAMETER DETAILS

As described earlier we chose a bin size of 4 and safety horizon $H_s = 80$ for *PointGoal1* and *PointButton1* environments and a bin size of 4 and safety horizon $H_s = 40$ for all the other environments. The batch size for training the S2C model was chosen between 512 and 5000 and we found that a batch size of 5000 led to better performance in Safety Gym environments and 512 for all the other environments. Additionally, we optimize for hyperparameters like when to update the S2C model $update - freq$ which was set to 100 for on-policy baselines and 20000 for off-policy baselines. Additionally, we used a learning rate of $1e - 6$ or $1e - 5$ for on-policy experiments and a learning rate of $1e - 3$ for off-policy baselines. The well-optimized hyperparameters of the baseline algorithms can be found in open-source implementations such as Omnisafe Ji et al. (2024)[5] (for CPO, CRPO, TRPO-PID, SauteRL) and FSRL Liu et al. (2024)[6] (for CVPO). For CSC Bharadhwaj et al. (2020) we used the hyperparameters specified in the original paper. We faced difficulty in training on-policy baseline algorithms on *SafeMetaDrive* with default hyperparameters from Omnisafe, fine-tuning hyperparameters revealed that lower target-kl stabilizes the baseline algorithms. $target - kl = 0.0005$. Additionally, we faced difficulty in stabilizing CVPO Liu et al. (2022) on *AdroitHandPen* and *Ant* tasks with the default hyperparameters. We couldn't stabilize the training with a limited hyperparameter search. The S2C model has the same network architecture as the policy which in most cases is an MLP with two hidden layers of size 64.

### A.4 ENVIRONMENTAL DETAILS

- *Island Navigation (Leike et al., 2017)* is a grid-world environment explicitly designed for evaluating safe exploration algorithms. The environment consists of an agent marked $A$, trying to navigate an island to reach a goal $G$ while avoiding water cells (marked in blue). Entering a water cell is considered a failure and the agent receives a heavy reward penalty along with episode termination. The state observation consists of the image of the island at a given time (as such the state is completely observable). The action space is $["left", "right", "up", "down"]$, so there is no action to stay in place. Because its a deterministic environment with a small distance between the start state and goal state, its not difficult for RL agents to just remember the action sequences instead of learning an accurate state-conditioned value function. In order to avoid such a situation, we instead created 4 copies of the same environment with different positions of start state and goal state as well as the location of the water cells (Fig. 11).

---

[5]https://github.com/PKU-Alignment/omnisafe
[6]https://github.com/liuzuxin/FSRL

- *AdroitHandPen (Rajeswaran et al., 2017)* is a manipulation environment, where a 24-degree of freedom Shadow Hand agent is learning to manipulate a pen from a randomly initialized start orientation to a randomly specified goal orientation. The input to the RL agent is the angular positions of the finger joints, the pose of the palm of the hand, as well as the current pose of the pen and the target pose of the pen. The action space is continuous and are the absolute angular position of the hand joints (24-dimensional). The pen falling on the ground is considered a failure and induces a cost. Since the cost-inducing state is also non-ergodic (irrecoverable), maximum cost for an episode cannot be greater than 1. The agent is provided a dense reward based on the similarity of the current pose to the target pose of the pen.

- *SafeMetaDrive (Li et al., 2022)* is an autonomous driving environment, where an agent is tasked with navigating the traffic containing static and dynamic participants. The agent has to frequently change lanes or apply brakes to avoid incurring costs. Any collision induces cost and the cost-limit is set to 1. The observation space consists of LiDAR information about the surrounding objects, ego-vehicle's pose, and safety indicators along with route points. The agent is rewarded for following the waypoints/route as well as for keeping the lane and for driving at high velocity. The action space consists of steering, brake and throttle values.

- *Safety-Gym (Ray et al., 2019)* is a navigation environment, we evaluate *SRPL* on *PointGoal1* and *PointButton1* tasks. In *PointGoal*, a randomly initialized point agent is trying to navigate to a randomly sampled goal position while avoiding hazards in the environment which are the cost-inducing states, so that the total cost for an episode doesn't exceed 10. In *PointButton1*, the agent is tasked with pressing the specific orange button, pressing the wrong button induces a cost and there are hazards as well as dynamic obstacles in the environment which are cost-inducing. Cost threshold is set to 10. In the original version of safety gym, the agent's observation consists of LiDAR information about every object including hazards, buttons and goal position individually. This prevents us from doing the generalization experiment since the dimensionality of the observation for *PointGoal1* and *PointButton1* are different because they have different objects in the environment. To address this we aggregate the Lidar information for all objects into one LiDAR observation that for every LiDAR point provides the distance to the nearest object as well as the corresponding object label. The action space for the point agent is the two-dimensional (force applied to move the point agent and the velocity about the z-axis). The reward function for the agent is based on the distance to the goal state.

- *Mujoco environments (Todorov et al., 2012):* We evaluate *SRPL* on three locomotion tasks (*Ant*, *Hopper* and *Walker2d*). The goal in all three tasks is to run as fast as possible while not falling on the ground. Falling on the ground is considered a failure and induces a cost. We treat the agent fallen on the ground as a non-ergodic state and thus the maximum cumulative cost for an episode can be 1. The agent's state input is the joint position and velocity of the robot's body parts as the centre of mass-based external forces acting on the body. The action space is the torques applied to each joints. The agent's reward is proportional to the velocity of the agent, and the agent is penalized for applying high torque on its joints.

## A.5 ADDITIONAL RESULTS

In addition to the results presented in the main paper, we perform experiments on Mujoco environments (*Ant*, *Hopper* and *Walker2d*) and show results for CPO (Achiam et al., 2017) and CRPO (Xu et al., 2021) along with their SRPL counterparts. From Fig. 12, we can see for the *Ant* environment, CRPO significantly outperforms CPO in terms of task performance (i.e. return) but also leads to more failures during training. SR-CRPO has similar performance in terms of return but significantly lower total failures during training thus making the algorithm safer. On the other hand, SR-CPO leads to an improvement in the performance in terms of return in comparison to CPO, while incurring similar but fewer failures. Similarly, for *Hopper* SR-CRPO outperforms all other baselines in terms of task performance as well as leads to the least total failures during training. SR-CPO also shows improvement over CPO in terms of safety by incurring fewer failures during learning. Finally, on *Walker2d*, SR-CPO outperforms all the baselines in terms of task performance while incurring the same number of failures as SR-CRPO. We observe a consistent improvement in either the safety or efficiency of the SRPL versions of the algorithms in comparison to their vanilla counterparts.

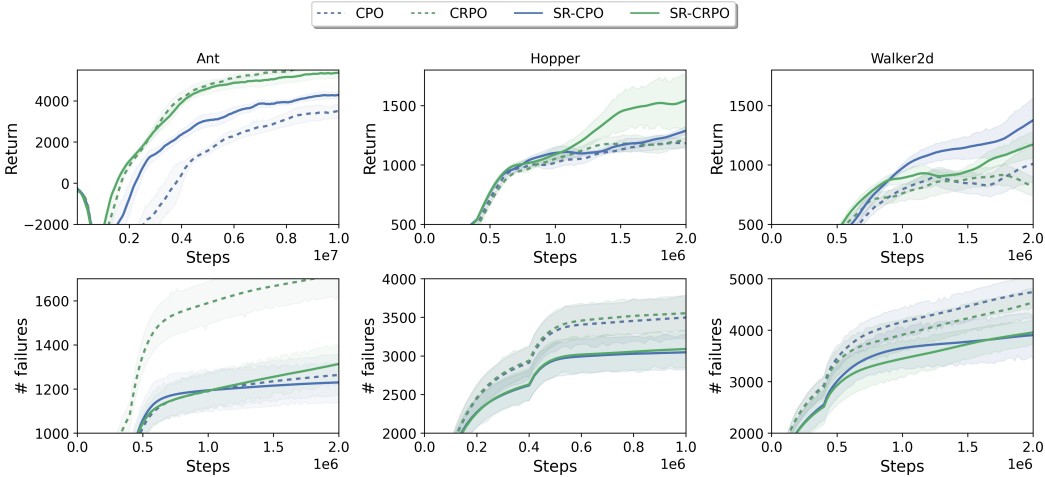

Figure 12: We report the performance of safety-informed SRPL agents (denoted SR-*) on Mujoco environments (Ant, Hopper and Walker2d). For these experiments, both the S2C model and the policy have been randomly initialized so no prior information has been provided to the agent. Safety-informed agents consistently outperform their vanilla counterparts on both safety during learning as well as sample efficiency. Results were obtained by averaging the training runs across five seeds. The input to the RL agent is the joint state and velocity.

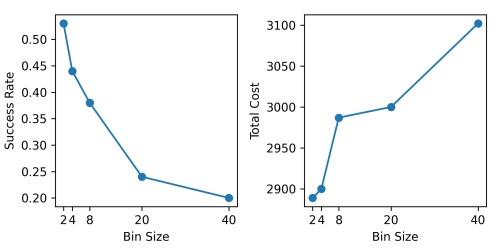

Figure 13: *Choice of bin size:* Results on SafeMetaDrive show that lowering the bin size better the performance of the model.

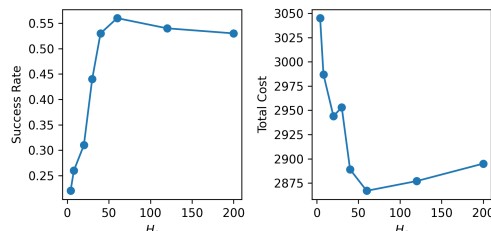

Figure 14: *Choice of safety horizon $H_s$* Results on MetaDrive show that higher safety horizon $H_s$ leads to improvement in performance but above a threshold the curve plateaus.

To provide more clarity into the results presented in the paper, we have added Fig. 16, which highlights the episodic cost for AdroitHandPen and SafeMetaDrive environments as well as the respective cost-limits. In our experiments we observed that setting the cost-limit to 0 for AdroitHandPen or the Mujoco environments was leading to the policy failing to learn anything, so we set the cost-limit to 0.1 ($\beta = 0.1$) for AdroitHandPen as well as Mujoco experiments (Fig. 12). For clarity we've also added pairwise plots for baseline algorithms with their SRPL versions on *AdroitHandPen* Fig. 17, *SafeMetaDrive* Fig. 18, *PointGoal1* Fig. 19 and *PointButton1* Fig. 20.

## A.6 ADDITIONAL ABLATIONS

To further analyze the safety representations proposed in the paper, we perform additional ablations that study the effect of design choices like safety horizon ($H_s$) and bin size for learned distribution. We also study the generalization ability of the learned safety representations across constraint thresholds.

### A.6.1 EFFECT OF SAFETY HORIZON AND BIN SIZE

In Fig. 13 and 14 we study the effect of different choices of bin size and safety horizon on the performance of the policy both in terms of safety (in terms of total cost) and efficiency (in terms of success rate) for *SafeMetaDrive* environment. In Fig. 14, we analyze the effect of varying safety horizons for a fixed bin size of $bin - size = 2$. We can see that both success rate and total cost almost plateau post the bin size of around 50, the dimensionality of the safety representation can be

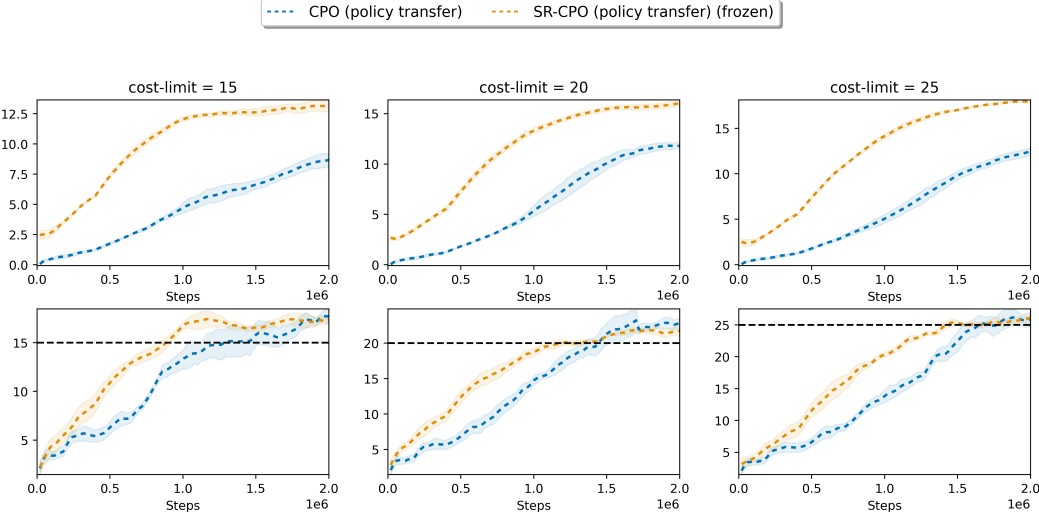

Figure 15: *Generalization across Cost thresholds*: Since safety representations are learned to model the proximity to unsafe states, they can be generalized across cost thresholds. Here we show that safety representation learnt on *PointButton1* with cost-limit = 10 can be transferred to *PointGoal1* task for cost-limits other than 10 without the need for fine-tuning and results in improved sample efficiency.

the cause behind the slight deterioration in performance beyond $H_s = 50$. Fig. 13 studies the effect of bin sizes given a fixed safety horizon $H_s = 40$, from the figure it is clear that smaller bin sizes lead to better performance both in terms of safety and efficiency of the SRPL agents.

## A.6.2   GENERALIZATION OF SAFETY REPRESENTATION ACROSS COST THRESHOLDS

Instead of modelling the likelihood of constraint violation, safety representation encodes the distribution over steps to cost or distance to cost-inducing/unsafe states. This design choice is based on the principle that we are interested in modelling state-centric features that are generalizable and don't depend on the task definition (for e.g., the cost threshold definition). We perform experiments on *PointGoal1* environment for the transferability of safety representation across constraint thresholds. For this experiment, we mimic the structure of the generalization experiments shown in Sec. 5.3. We treat *PointButton1* as the source task and *PointGoal1* as the target task where the safety representation as well as policy for both CPO and SR-CPO is trained on *PointButton1* task. In order to study the generalizability of safety representations across cross thresholds we don't fine-tune the safety representation on the target task for different cost thresholds thus freezing the *S2C* model. From Fig. 15, we can see that SR-CPO (policy transfer) (frozen safety) significantly outperforms CPO (policy transfer), thus transferring the policy and frozen safety representation across tasks enables SRPL agents to learn more samples efficiently while ensuring constraint satisfaction for different constraint thresholds.

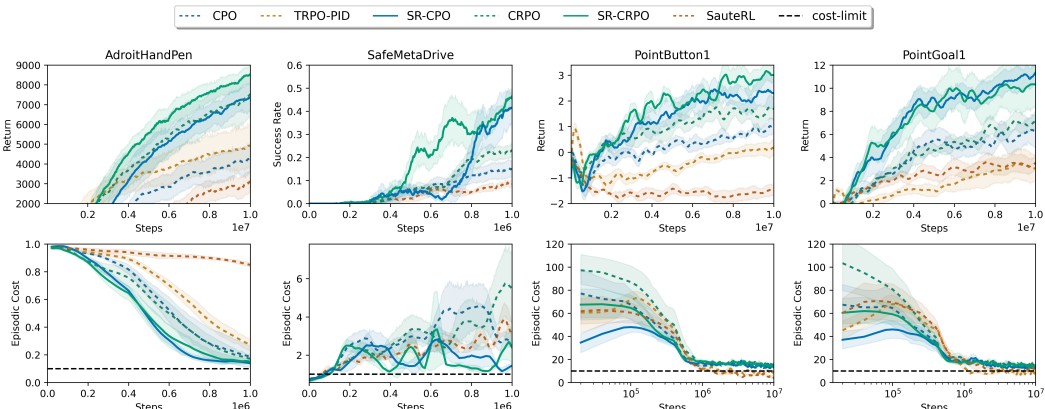

Figure 16: The figure shows Episodic Cost / Episodic Failure for AdroitHandPen and SafeMetaDrive experiments

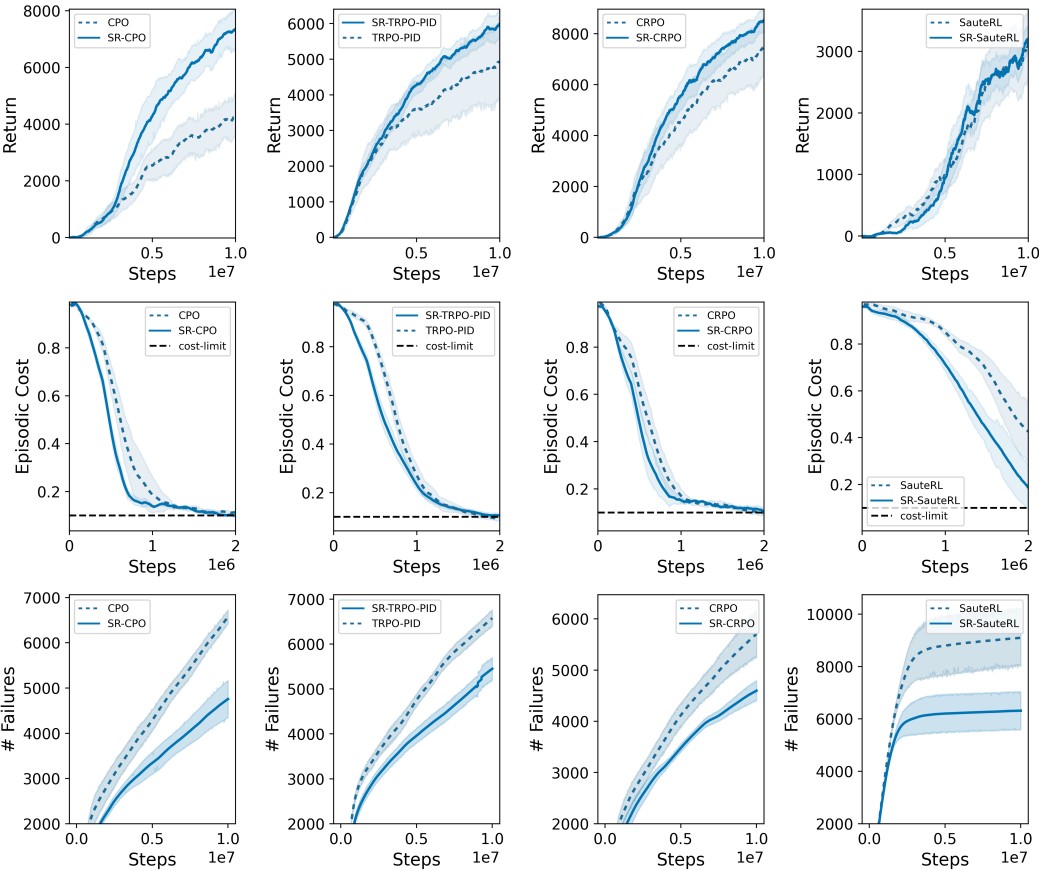

Figure 17: *AdroitHandPen*: We plot all the baselines and their SRPL counterparts

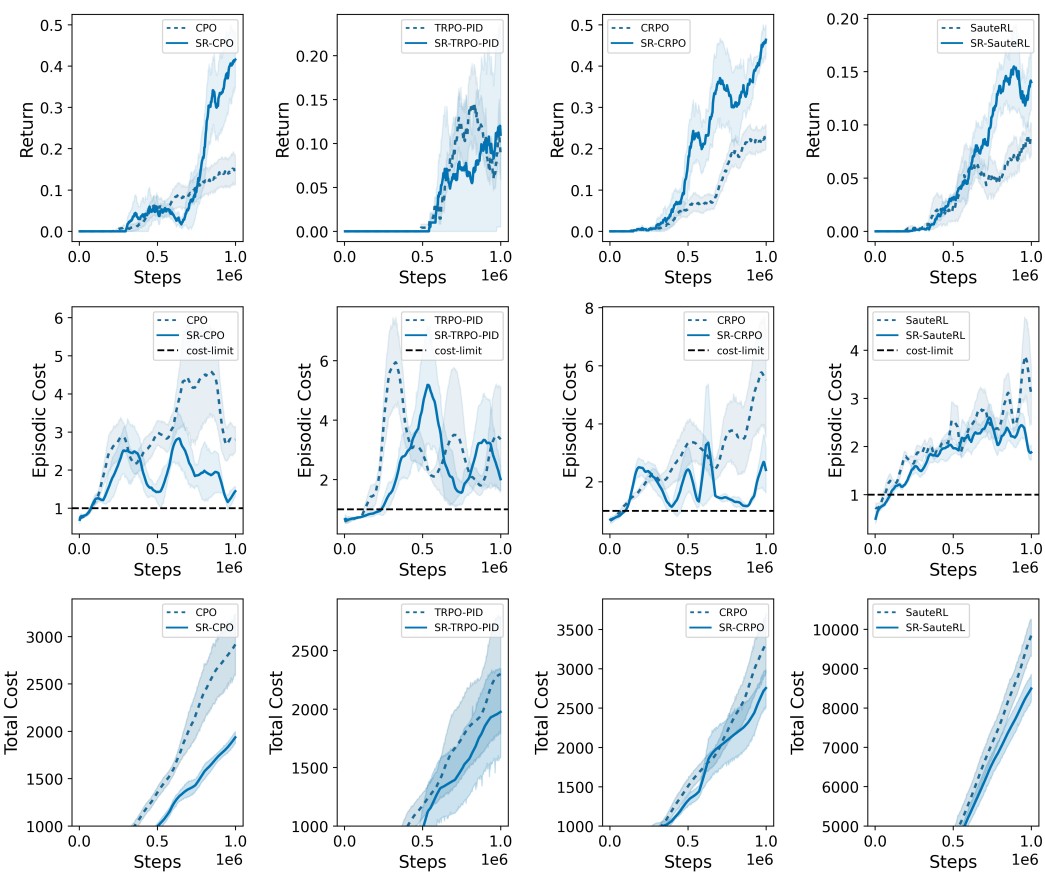

Figure 18: *SafeMetaDrive*: We plot all the baselines and their SRPL counterparts

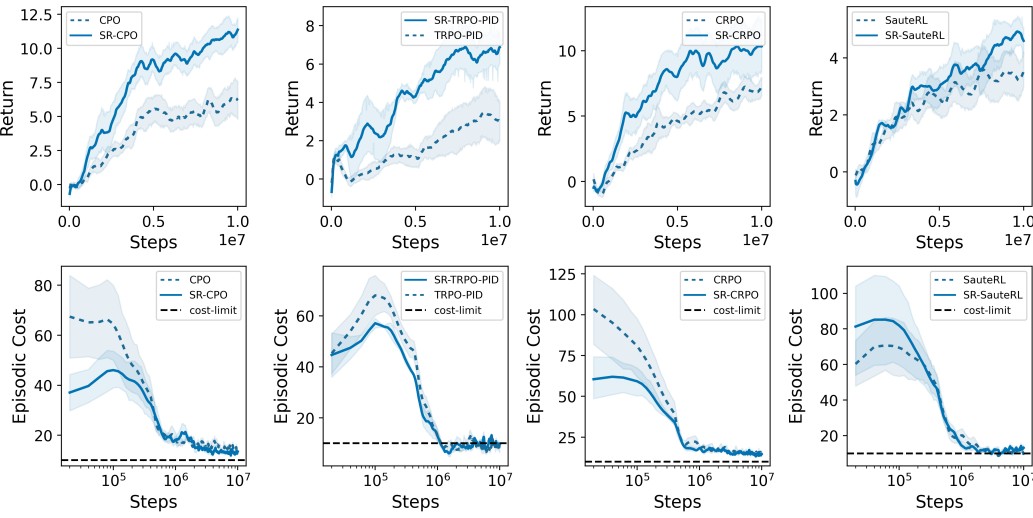

Figure 19: *PointGoal1*: We plot all the baselines and their SRPL counterparts

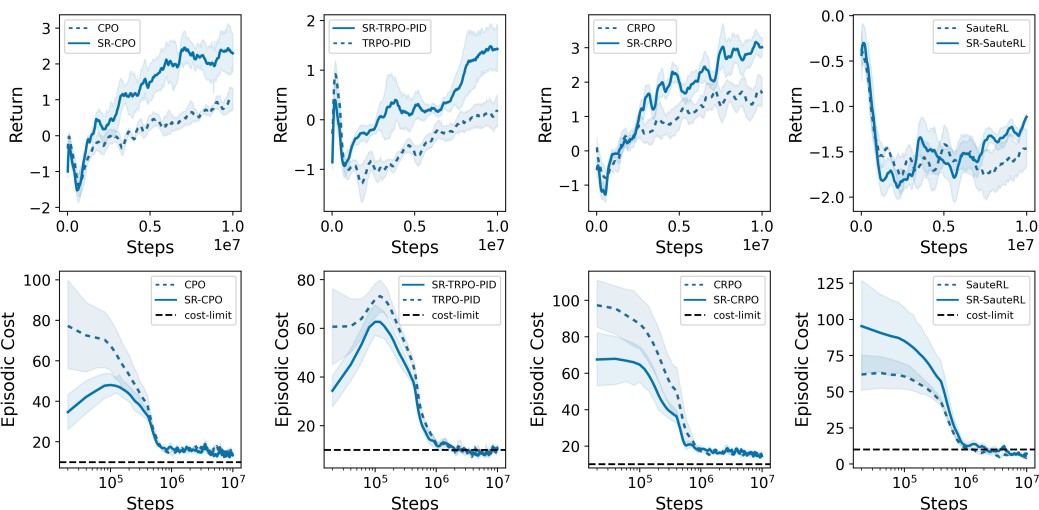

Figure 20: *PointButton1*: We plot all the baselines and their SRPL counterparts

