# OpenReview forum: "Safety Representations for Safer Policy Learning"
_ICLR.cc/2025/Conference — ICLR 2025 Poster_

### Official Review · Reviewer_DjQM · 2024-10-19

**Soundness:** 2
**Presentation:** 2
**Contribution:** 2
**Rating:** 5
**Confidence:** 4

**Summary:**

In this paper, the authors propose a new reinforcement learning framework with a specific focus on "safe exploration." The proposed framework, named RIPL, can be described as a data-augmentation method that introduces additional risk-related representations into the original state-space, leading to more efficient safe policy learning. The idea is intriguing, and the toy experiments illustrate the motivation behind it well. However, the paper faces some issues with clarity. Additionally, the experimental results are limited and not sufficiently convincing. I hope the authors can address the concerns I have outlined below.

**Strengths:**

1. The idea presented in this paper is interesting, and the implementation looks simple and practical.
2. The proposed framework, RIPL, is flexible and can easily be combined with many existing online safe RL algorithms.
3. The toy experiments do a good job of showing the motivation behind the approach.

**Weaknesses:**

1. As the authors describe, $R_H(s)$ represents the probability that the agent will remain safe for the entire episode. However, this probability is highly dependent on the policy being used. In Appendix A, it is derived based on the specific policy, which raises a concern: how is the risk buffer collected and updated to train $R_H(s)$? If all new trajectories are collected without discarding old ones, the estimation will be inaccurate. On the other hand, if only on-policy trajectories are used, the approach is not safe and requires a large number of samples for convergence.
2. A similar idea of learning risk-related value estimation has been proposed in [1], with several works following this approach to learn Safe Q via a safe-Bellman operator. This method offers more solid theoretical guarantees.
3. The caption for Figure 3 is unclear and does not align with the figure, which could cause confusion.
4. The baseline algorithms used in the experiments seem somewhat outdated, and the number of experimental results provided is too limited to fully demonstrate the advantages of the proposed algorithm.

[1] Fisac, Jaime F., et al. "Bridging hamilton-jacobi safety analysis and reinforcement learning." 2019 International Conference on Robotics and Automation (ICRA). IEEE, 2019.

**Questions:**

1. In this paper, the authors frequently claim that the "RL agent overfits the negative samples." However, it is unclear whether this "overfitting" refers to the representations, Q-functions, or actions. In Column 1 of Figure 1, how does the training curve reveal this "overfitting problem"? Could the authors provide a more detailed explanation, along with additional evidence?

2. In the experiments, are the environments image-based or state-based? Since all environments provide both versions, it would be helpful to clarify this in the figure/table captions.

3. The experimental results provided are too limited to convincingly demonstrate the algorithm's effectiveness. Could the authors present more results from Mujoco (hopper, halfcheetah, walker2d), Adroit (hammer, door), and Safe Gym (Point/Car/Doggo goal/button/push 1/2)? If possible, including results from MetaDrive, which is arguably the most challenging environment for testing algorithms with safety considerations, would strengthen the case.

4. Given that the focus of this work is "Safe Exploration," it would be beneficial to emphasize the safety-related results during the exploration process. Could the authors offer more insights on this aspect?

5. The key point for RIPL is the risk buffer, but it is not adequately discussed in the main text. How is the buffer updated? A very large number is assigned to the risk-batch-size. Does this imply that it never updates and contains all information throughout the training process? If so, the risk-distance probability estimation must be inaccurate since the dataset's policies vary significantly. The estimation should differ substantially for different policies.

6. Since $R_H(s)$ is a distribution, its dimension varies significantly across environments with different horizons. What happens if the total horizon timestep is extremely large? In most cases, we consider infinite-horizon MDPs. How can RIPL generalize to that?

---

> ### Author Response · Authors · 2024-11-28
> **Response to Reviewer DjQM**
>
> We sincerely thank the reviewer for recognizing the practicality and flexibility of our approach and for helping identify inconsistencies in the paper. We believe we have addressed all of your concerns in the manuscript with the changes highlighted in blue for ease of reference, but we remain available to address any further questions or comments before the end of the discussion period.
>
> **[W1]:  Imprecise definition of risk representation (now safety representation).**
>
> 1. Thanks to the reviewer’s comment, we’ve identified that this specific description of $\( R_H(s) \)$ (now $\( \mathfrak{S}_{H}(s) \)$) was imprecise. We’ve corrected this in the revised paper by replacing the line with: “$\( \mathfrak{S}_H (s) \)$ represents the probability that the agent remained safe throughout the fixed safety horizon $\( H_s << H\)$ without encountering an unsafe state based on the agent’s past experience.” As such, the safety representation encodes information about the agent’s past experience and cannot accurately predict whether the agent will fail following a novel policy.
>
> 2. We agree with the reviewer’s observation regarding the tradeoff between on-policy and off-policy data for learning the safety representations. Following common RL practice, we store all policy rollouts in a queue within a replay buffer, prioritizing recent policies. Specifically, we use a 1M-size buffer for all environments bar SafeMetaDrive where the buffer size is even smaller (200K), removing old trajectories in a FIFO manner. Further details are provided in Sec. 3.3 and Appendix A.1. The effect of modelling the safety distribution with off-policy vs on-policy data is explored in Fig. 8.
>
> 3. We have removed the theoretical analysis (previously in Appendix A) as it did not contribute relevant insights to the paper and could potentially confuse the reader.
>
> 4. In response to the reviewer’s feedback, we have added the necessary implementation details and design discussions in Sec. 3.3 and Appendix A.1, including a dedicated discussion on design choices in Appendix A.1.1.
>
>
> **[W2]: Safe Bellman Operator Based Methods**
>
> We thank the reviewer for pointing out this reference, which we have now included in the Related Work section. Safe Bellman operator-based methods, such as those in [4], CSC [5], and SQRL [6], also aim to model the likelihood of agent failure. However, unlike our approach, these methods use this information to constrain policy learning rather than informing it. In contrast to [4, 5, 6], we do not model a scalar probability of failure; instead, we learn a distributional representation of the state, characterizing it as a distribution over the proximity to unsafe states based on the agent’s past experience.
> Results for CSC [5], a popular off-policy baseline that uses the Safe Bellman operator to model the likelihood of failure for constraining policy actions, show that such methods can lead to overly conservative policy learning (Fig. 5 in the revised paper). Additionally, our experiments demonstrate that the proposed method, SRPL, significantly improves the performance of CSC [5], both in terms of constraint satisfaction and sample efficiency (Fig. 5).
>
> Being a representation learning approach, our method indeed provides no new theoretical guarantees, but it does inherit the theoretical guarantees of the baseline method to which it is applied.
>
> **[W3]:** Thanks for pointing this out. We’ve corrected the caption for Figure 3.
>
> **[W4]: Additional Baselines and More Environments**
>
> 1. Taking the reviewer’s comment about limited and outdated baselines into consideration, we’ve added three additional baselines from recent online SafeRL literature: CRPO [1], CVPO [3], and SauteRL [2]. The results for these baselines have been shown in Fig. 4 & 5, for the sake of better clarity we’ve included pairwise plots in Fig. 18-21 in the Appendix.
>
> 2. In the original paper, our focus was to show the effectiveness of our method across a diversity of robotics tasks. We’ve since added results for three more environments: SafeMetaDrive, Hopper and Walker2d. The importance of SafeMetaDrive as pointed out by the reviewer in [Q3] convinced us to put the results for SafeMetaDrive in the main paper (Fig. 4) and move the results for Ant to the Appendix (Fig. 12).
>
> We'll continue in the next comment.
>
> [1] Xu et al. CRPO: A New Approach for Safe Reinforcement Learning with Convergence Guarantee In ICML 2021
>
> [2] Sootla et al. Sauté RL: Almost Surely Safe Reinforcement Learning Using State Augmentation In ICML 2022
>
> [3] Liu et al. Constrained Variational Policy Optimization for Safe Reinforcement Learning In ICML 2022
>
> [4] Fisac, Jaime F., et al. "Bridging hamilton-jacobi safety analysis and reinforcement learning." 2019 International Conference on Robotics and Automation (ICRA). IEEE, 2019.
>
> [5] Bhardwaj et al. Conservative Safety Critics for Exploration In ICLR 2021
>
> [6] Srinivasan et al. Learning to be Safe: Deep RL with a Safety Critic

---

> > ### Author Response · Authors · 2024-11-28
> > **Continued Response to Reviewer DjQM**
> >
> > **[Q1]: Overfitting and Primacy Bias in Safe RL**
> >
> > We have realized that the phrase “RL agents overfit on negative experiences” oversimplifies the cause of conservatism in Safe RL. The main issue is primacy bias, where agents overfit to early experiences due to limited data diversity from poor state-space exploration, as discussed in [7]. This overfitting compounds: the agent collects suboptimal data, further degrading learning efficiency and limiting generalization. In safety-critical settings, agents often violate constraints early on, incurring severe penalties that lead to overly conservative policies, overestimating risk and restricting exploration.
> > As shown in Fig. 1 (col. 1), initial exploration phases lead to high failure rates for all agents. In the absence of accurate safety information, agents converge to suboptimal policies that avoid unsafe states, leading to low returns. Early failures, penalized by negative rewards, result in low Q-value estimates (Fig. 1, row 1, col. 2), causing the agent to reduce exploration and oscillate between states to minimize risk (Fig. 1, row 1, col. 3). This conservative strategy demonstrates the agent's reliance on memorized actions rather than adaptive learning, prioritizing safety at the cost of optimality. We’ve also included additional experiments and discussions into the primacy bias and overfitting Safe RL in Sec A.1. of the appendix. For additional insights into overfitting and primacy bias, we recommend [7], which offers an empirical analysis of this phenomenon.
> >
> > **[Q2]: Specifying the observation space**
> >
> > We thank the reviewer for pointing out the missing information regarding the observation space. We have now included these details in the captions of Fig. 4, 5, and Table 1, as suggested. The observation space for all results, except those in Table 2, is state-based. In Table 2, we investigate the effect of the safety representation for higher-dimensional sensor modalities, such as RGB and Depth.
> >
> > **[Q3]: Limited Experimental Results and Baselines**
> >
> > Thanks to the reviewer’s comment we have added more baselines and added three more environments: SafeMetaDrive, Hopper, and Walker2d to the paper. We’ve discussed this in [W3].
> >
> > **[Q4]: Emphasis on Safety-related results during exploration**
> >
> > Since most Safe RL algorithms are built on top of TRPO, PPO, or SAC, which use stochastic policies with entropy-based regularization, exploration and exploitation are inherently intertwined. This makes decoupling exploration and exploitation, as is possible with value-based methods like DQN, more challenging. In such settings, metrics like total failures, total cost during training, or early constraint satisfaction provide effective proxies for evaluating the agent’s ability to explore safely. These metrics are widely used in the Safe Exploration literature [1][2][3][5][6]. Table 1 demonstrates that SRPL improves the performance of baseline algorithms by enabling safer exploration, as evidenced by fewer failures and lower costs during training, leading to better overall performance.
> >
> >
> >
> > **[Q5]: Insufficient Implementation Details**
> >
> > 1. We agree with the reviewer that the original paper lacked sufficient discussion on the replay buffer update. To address this, we’ve added an "Implementation Details" section in Sec. 3.3 and expanded on it in Sec. A.2 of the appendix, covering design choices and buffer updates. Additional details are provided in [W1] (b).
> > 2. The risk batch size varies across environments: larger batches improve SRPL performance in Safety Gym, while smaller batches are better for SafeMetaDrive. The replay buffer is $200K$ for SafeMetaDrive and $1M$ for other environments. Initially, the risk batch size stores all samples, but this changes as training progresses. As described in [W1], we update the buffer by discarding older policy rollouts in a FIFO manner to ensure the relevance of safety representations for the current policy.
> >
> > **[Q6]: Dimensionality of Safety Distribution and Safety Horizon $H_s$**
> >
> > Thank you for the comment. Instead of learning the safety distribution over the entire time horizon, which would be infeasible for large or infinite horizon MDPs, we model it over a fixed safety horizon $H_s$. We acknowledge that this detail was missing in the original version of the paper. This has been clarified in Sec. 3.2 of the revised version. Specifically, for Safety Gym environments, the safety horizon is set to 80 with a bin size of 4, while for all other experiments, the safety horizon is fixed at 40 with a bin size of 2. This approach enables generalization to both finite and infinite horizon MDPs. Additional implementation and hyperparameter details can be found in Sec. A.2 of the appendix.
> >
> > Relevant citations in the last comment. Not putting it again because of character limit
> >
> > [7] Nikishin el al. The Primacy Bias in Deep Reinforcement Learning In ICML 2022

---

> > > ### Comment · Reviewer_DjQM · 2024-11-28
> > >
> > > I must acknowledge that the authors have invested considerable effort during the rebuttal phase. It appears that they have taken all suggestions into account, resulting in a comprehensive rewrite of the paper. Consequently, I am inclined to revise my score to 5.
> > >
> > > However, this should not be interpreted as an endorsement for accepting the paper for the conference. The original version had a **significant technical issue** that, regrettably, went unnoticed by the other reviewers who provided unduly high scores. In the revised manuscript, the authors have adopted an intuitive approach to address what was originally a theoretical misconception. The theoretical analysis has been deleted, and the experiments has been entirely redone. I believe that the revised paper should be re-evaluated by at least two additional professional reviewers before it got accepted.
> > >
> > > I am merely offering my perspective on the matter. The final decision should be deferred to the Area Chairs.

---

> > > > ### Author Response · Authors · 2024-11-28
> > > >
> > > > We respectfully disagree with the reviewer that there was a significant technical issue in the original manuscript. We certainly concede that the use of the term risk induced confusion, and that there was a lack of clarity surrounding how the S2C (formerly risk) model was trained. We have removed the theoretical analyses in the Appendix because we found that they were not adding significant value over and above the empirical results as pointed out by Reviewer jX1B. The experiments have not been redone entirely but results on new environments and additional baselines have been added according to the reviewers' suggestions.

---

### Official Review · Reviewer_jX1B · 2024-10-31

**Soundness:** 3
**Presentation:** 3
**Contribution:** 2
**Rating:** 6
**Confidence:** 5

**Summary:**

The paper proposes a safe RL method based on augmenting the state space with a learned safety representation, which is referred to as Risk-Informed Policy Learning. This state-conditioned representation is learned during training, and captures the probability of safety violations in each of the next several timesteps. The goal of incorporating this representation is to learn policies that satisfy safety constraints without being overly conservative in terms of task performance. Experiments across 4 different tasks evaluate task performance and safety metrics, where the learned state representation is combined with several different safe RL algorithms.

**Strengths:**

- **[S1] Important topic:** Safe / Constrained RL is an important topic, and training policies that satisfy safety constraints without being overly conservative is an important goal.
- **[S2] Compatible with many RL algorithms:** The proposed method augments the state with a learned representation, so it can be applied to many different RL algorithms. This is also demonstrated in the experiments, where it is applied to 4 different safe RL algorithms.
- **[S3] Experiments across diverse set of tasks and several base algorithms:** Experiments consider manipulation, locomotion, and navigation tasks, and the proposed state representation is combined with 4 different safe RL algorithms. Additional ablation analyses are also included, which consider different input modalities and alternate choices for the learned safety representation.

**Weaknesses:**

**[W1] Confusing use of the term “risk” throughout the paper**
- A risk measure is a function that maps a random variable to a scalar value (e.g., CVaR). Risk-sensitive / risk-aware / risk-averse RL methods typically apply a risk measure over a distribution of future returns. This is very different from the ideas proposed in this work, where a “risk function” is mapping states to a distribution over timesteps. I strongly suggest that the authors change the term “risk-informed” to something different that better reflects what is being proposed in this paper.

**[W2] The definition of the risk model is inconsistent across the paper**
- The risk model is defined as a probability distribution over the number of steps to failure (i.e., constraint violation in (2)) in several places (lines 60-61, 172-173, 212, 256-257). I believe this formulation makes the most sense, as it captures information about constraint violation similar to the state augmentation proposed in [1]. However, in other places (lines 221-226, 461, 719-720, 791-792) the risk model is described as a probability distribution over the number of steps to an unsafe state (i.e., a state where $c=1$ but not necessarily a constraint violation).

**[W3] Experimental analysis is not clearly presented**
- The safety metrics presented in the Experiments section are difficult to interpret. I cannot tell what percentage of trajectories end in safety violations in the AdroitHandPen and Ant tasks based on the #failures metric, and I cannot tell whether the safety constraint is satisfied in PointGoal1 and PointButton1 tasks based on the Cost-rate metric. In Figure 5, it does not look like the safety constraint of $\beta=10$ is satisfied for these tasks. See [Q1].
- There is not enough detail on how the safety representation is implemented in the Experiments. There might be some information in Appendix A.6, but it is difficult to understand. See [Q1].
- I do not understand how the results in Figure 7 support the claims about transferability. Policy transfer works well both with and without the RI representation, and it does not look like transferring the risk model leads to better performance compared to just training RI-CPO from scratch as shown in Figure 5. See [Q1].
- I do not understand some of the presentation choices in Figures 5-7: returns and costs are shown for different numbers of total timesteps; some RI algorithms are shown while others are not; some algorithms show performance throughout training while others are just horizontal lines.

**[W4] Appendix is not well-written and does not present significant results**
- The results in the Appendix rely on unusual assumptions and do not provide significant insights into the main problem considered in the paper.
- Proposition 0.3 is not new. See Theorem 3 in [2].
- Assumptions / proofs are often not explained in an appropriate amount of detail: the definition of $r(s)$ in Proposition 0.1 does not seem to be directly used in the proof; $\eta$ introduced in lines 928-929 does not appear to be used, as (19) looks to just be a repeat of (18).
- Several incorrect, hardcoded equation references. Strange numbering of propositions.

---

**References:**

[1] Sootla et al. Sauté RL: Almost Surely Safe Reinforcement Learning Using State Augmentation. In ICML 2022.

[2] Syed et al. Apprenticeship Learning Using Linear Programming. In ICML 2008.

**Questions:**

**[Q1] Experiment clarifications**
- Please provide the average total cost per trajectory for each experiment, and how this relates to the safety constraint budget $\beta$.
- Please describe the implementation of the risk model in more detail. What horizon is used? What network architecture is used? Other key details?
- Could you please provide more detail on how Figure 7 (and the RI-CPO training from scratch in Figure 5) support the claim about transferability? I also think that RI-CPO should be included in Figure 7 to more easily compare against it without needing to look back to Figure 5.

**[Q2] Choice of RL algorithms**
- Why is CPO used as the main safe RL algorithm for most of the experiments? More recent safe RL algorithms have been shown to outperform CPO, such as CVPO [3] and LAMBDA [4].
- The approach in [1] also seems relevant to this work, since it incorporates information about safety constraint violation through state augmentation (this work considers a more detailed, learned state augmentation while [1] uses the remaining budget).

**Minor:**
- cost function notation is inconsistent: $c$ vs. $C$ vs. $\mathcal{C}$
- an index variable $i$ appears in (2) but is not used anywhere
- incorrect column / row references in lines 191-192
- floating quotation mark in the top left of Figure 3
- colors are not defined in Figures 3 and 9
- incorrect, hardcoded equations throughout Appendix
---

**References:**

[3] Liu et al. Constrained Variational Policy Optimization for Safe Reinforcement Learning. In ICML 2022.

[4] As et al. Constrained Policy Optimization via Bayesian World Models. In ICLR 2022.

---

> ### Author Response · Authors · 2024-11-28
> **Response to Reviewer jX1B**
>
> We would like to express our sincere gratitude for the thorough and detailed review of our work. We believe that the clarity and quality of the paper have been significantly enhanced by incorporating your valuable suggestions. We have addressed all of your concerns in the revised version, with the changes highlighted in blue for ease of reference.
>
> **[W1]: Confusing use of the term “risk” throughout the paper**
>
> We acknowledge the reviewer’s concern regarding the potential ambiguity of the term "risk," which is often interpreted as a risk measure. To address this, we have revised the terminology throughout the paper and method. Specifically, "risk representation" has been updated to "safety representation," and "risk model" is now referred to as the "Steps-to-Cost (S2C) model." Additionally, the framework has been renamed to "Safety Representation for Policy Learning (SRPL)" in the revised manuscript.
>
> **[W2]: The definition of the risk model is inconsistent across the paper**
>
> We agree with the reviewer about the inconsistency of the definition of the risk model in the original version of the paper. In our approach, we model the distance to unsafe states, which may not always correspond to a constraint violation or failure. We agree that modelling the distance to constraint violation is also a sensible choice. To address this in the paper, we’ve made a single definition of the S2C model (earlier risk model) in Sec. 3.2 and 3.3. Our design choice is based on our intuition of a state-centric safety representation that is independent of task specifications, enabling generalization across different tasks and constraints, as long as the definition of unsafe states remains consistent. In Appendix Figure 15, we present results demonstrating the transfer of the learned safety representation across tasks for different cost thresholds. Specifically, the original representation, trained with a safety threshold of $\beta=10$, was then frozen and applied to the target task for different cost thresholds.
>
> **[W3 & Q1]: Experimental Clarifications**
>
> Upon further reflection, we agree with the reviewer that the results were not presented clearly. The reasoning behind presentation choices like showing costs for the first $2M$ steps in the case of Safety Gym environments was to highlight the low episodic cost incurred by the SRPL (earlier RIPL) agents in comparison to the baseline algorithms. All algorithms converge around 2M timesteps in terms of Episodic cost. As a result, the early benefits of SRPL were difficult to see when plotted on 10M time steps. The choice behind not showing all of the SRPL (earlier RIPL) versions of the baseline algorithm was to prevent excessive overlap which was reducing the clarity of the figure. The same applies to some of the baselines plotted as horizontal lines.
>
> To address the reviewers’ concerns and improve the clarity of the paper, we’ve modified the presentation of the results in the following way:
>
> 1. We’ve plotted the results for Episodic Cost for Safety Gym with the x-axis on the log-scale in Fig. 4 thus showing the benefits of using SRPL on Episodic Cost clearly. We also now include a line indicating the constraint threshold for Safety Gym environments.
> 2. For AdroitHandPen and SafeMetaDrive environments, we’ve included a plot in the Appendix (Fig. 17) that shows the episodic cost along with the constraint threshold.
> 3. To show all versions of the vanilla algorithms and their SRPL counterparts, we’ve added pairwise plots for each environment and algorithm combination in Figures 18-21.
> 4. With the inclusion of CVPO (an off-policy algorithm) as a baseline, we decided to show off-policy algorithms (CVPO and CSC) with their SRPL counterparts separately in Fig. 5.
>
>
> We are almost at the character limit for this comment so we will continue with our response in a separate comment.

---

> > ### Author Response · Authors · 2024-11-28
> > **Continuation of Response to Reviewer jX1B**
> >
> > **[W3 & Q1]: Generalization of Safety Representations**
> >
> > 1. We acknowledge the reviewer’s comment regarding the lack of clarity in Figure 7 (now Figure 6). To address this, we have updated the figure with new results to better demonstrate the generalizability of the safety representation. Additionally, we have separated the results for safety transfer and policy transfer to enhance clarity.
> > 2. In the case of safety transfer, we show that frozen safety representations encode state-centric information, allowing for zero-shot transfer (without fine-tuning) to improve performance on the target task. While training SR-CPO directly on the target task yields more optimal performance, this is due to the differences between the PointGoal1 and PointButton1 environments, which share some objects but not others. We also demonstrate that fine-tuning the safety model on the target task recovers performance similar to SR-CPO while enabling faster constraint satisfaction.
> > 3. In the second experiment on policy transfer, we show that transferring both the policy and the safety representation results in significant improvements in sample efficiency. To emphasize this, we fine-tuned the policy for only 2 million timesteps and observed that even with frozen safety representations, the improvement in sample efficiency was substantial. Furthermore, fine-tuning both the safety representations and the policy led to even greater gains in sample efficiency, as well as improved constraint satisfaction.
> >
> > **[W4]: Appendix is not well-written and does not present significant results**
> >
> > We thank the reviewer for their valuable feedback. Indeed, we faced challenges in determining how best to incorporate the theoretical analysis presented in the Appendix of the original paper. While we initially believed that the theoretical results might offer additional insights into the functioning of the safety representations, upon further reflection, we agree with the reviewer that they were neither fully developed nor added significant value to the paper. Consequently, we have decided to remove the theoretical analysis entirely and instead focus on providing more experimental details, ablations, and additional results.
> >
> >
> > **Questions:**
> >
> > **[Q1]: Experimental Clarifications**
> >
> > Partly answered in [W3].
> >
> > We have provided a more comprehensive description of the implementation details in Section A.2 of the appendix. For the Safety Gym environments, the safety horizon $H_s$ was set to 80 timesteps, while for all other environments, it was fixed at 40 timesteps. The architecture of the safety model (now referred to as the S2C model) mirrors that of the policy model, with the key difference being the output layer, which corresponds to the dimensionality of the distribution determined by the safety horizon and bin size. For most experiments, we used two hidden layers, each with 64 units. Additional details on the training dynamics and practical considerations are also included in Section A.2 of the appendix.
> >
> > **[Q2]: Choice of baseline algorithms**
> >
> > In response to the reviewer’s comments, we have added results for additional baseline algorithms: CVPO [3], CRPO [1], and SauteRL [2]. As expected, CVPO demonstrates greater sample efficiency than CPO, and we have included results for CVPO alongside CSC (another off-policy algorithm) in Figure 5. We did not include Lambda [4] for two reasons: 1) Lambda is a model-based method, whereas all the methods presented in this paper are model-free, and 2) in the original version of the paper, Lambda uses visual input, while our Safety Gym environments utilize LiDAR.
> >
> > [1] Xu et al. CRPO: A New Approach for Safe Reinforcement Learning with Convergence Guarantee In ICML 2021
> >
> > [2] Sootla et al. Sauté RL: Almost Surely Safe Reinforcement Learning Using State Augmentation In ICML 2022
> >
> > [3] Liu et al. Constrained Variational Policy Optimization for Safe Reinforcement Learning In ICML 2022
> >
> > [4] As et al. Constrained Policy Optimization via Bayesian World Models. In ICLR 2022.
> >
> >
> > **Minor:**
> >
> > We would like to thank the reviewer for identifying these minor corrections that were overlooked during our proofreading. We have made the necessary changes to address them in the revised version of the paper.

---

> > > ### Author Response · Authors · 2024-12-02
> > > **Reviewer jX1B followup**
> > >
> > > Dear reviewer jX1B, Thank you once again for your thorough and constructive review. We have carefully incorporated your suggestions and addressed your questions in the revised manuscript and the comments provided above.
> > >
> > > As the discussion period draws to a close, we would like to confirm whether you have any remaining concerns. We would be glad to answer any further questions or sincerely appreciate any additional feedback you may have.

---

> > > > ### Comment · Reviewer_jX1B · 2024-12-02
> > > >
> > > > Thank you for your responses and revisions. My main comments have been addressed by the revised paper, and I have increased my score to reflect this. Please update the final version to satisfy the 10 page limit, fix typos (SRPL vs. SPRL, etc.), and update old references to the risk model (including in tables, figure legends, section headers).
> > > >
> > > > One follow-up question on the revised paper: in many experiments, it appears that several algorithms (including SR versions) do not learn to satisfy the safety budget at the end of training. This is a little concerning in a safe RL paper. Could you please comment on this?

---

> > > > > ### Author Response · Authors · 2024-12-02
> > > > > **Additional Clarifications**
> > > > >
> > > > > Thank you for your feedback. We will ensure a thorough proofreading of the paper and address the typos and page limit issues in the final version.
> > > > >
> > > > > Regarding constraint satisfaction, the episodic cost oscillates between satisfying and violating the constraints after 2M timesteps in the SafetyGym environments for both CPO and CRPO, which are based on trust region optimization to ensure constraint adherence. We have smoothed the curve for better visualization. This issue is not observed in Lagrange-based Safe RL methods, such as TRPO-PID, CSC, or penalty-based methods like SauteRL and their SR counterparts, as shown in Figures 4 and 18-21.
> > > > >
> > > > > The inability of CPO and CRPO to satisfy constraints exactly may stem from approximation errors in the linearization of the constraints, which may not fully capture the true non-linear constraint landscape. In our experiments, we tested various values for the backtracking line search steps to determine if higher values could resolve this issue but found no improvement. Since SRPL is a representation learning approach that does not modify the underlying algorithm’s objective or optimization strategy, it faces the same challenge. Although we did not have sufficient time to study CVPO in detail, we will provide additional insights on this in the final version of the paper.
> > > > >
> > > > > We also observed that in SafeMetaDrive, the episodic cost oscillates as the agent improves its performance, leading to more scenarios of constraint violation, followed by penalties that reduce the episodic cost. This pattern is consistent across all baseline methods we compared. When we enforce strict constraint adherence through heavy penalties, such as a high initial Lagrange multiplier in TRPO-PID, the agent fails to solve the task. We’ll strive to provide a more thorough analysis of the tradeoff between constraint violations and reward in the final version of the paper.

---

### Official Review · Reviewer_maKe · 2024-11-02

**Soundness:** 3
**Presentation:** 4
**Contribution:** 3
**Rating:** 8
**Confidence:** 4

**Summary:**

The main idea of the paper is to augment the state space with a distance to “unsafe” regions and use this augmented state-space to learn safe RL policy. The paper presents a number of experiments and comparisons to safe RL methods as well as an ablation study showcasing different aspects of the algorithm.

**Strengths:**

1. The paper is very well-written
2. The motivational example is great! I would recommend getting the message a bit more fleshed out. That is, that adding a distance measure in this example with sparse rewards made a huge difference for training. The paper tries to do the same trick in safe RL, where we often treat safety similarly to RL in the sparse reward setting - either safe or not safe.
3. The paper has a great ablation study

**Weaknesses:**

1. The CPO baseline is fine, but not amazing. I think it would be good to use at least one more baseline for demonstration. There are a few implementations in https://github.com/PKU-Alignment/safety-gymnasium
2. Ideally, it would be good to see more risk measure distances compared to the presented one.

**Questions:**

1. The paper
   > Jiang, H., Mai, T., Varakantham, P., & Hoang, M. H. (2023). Solving Richly Constrained Reinforcement Learning through State Augmentation and Reward Penalties. arXiv preprint arXiv:2301.11592.

is another example of state-space augmentation for safety.

2. The authors use a time-dependent risk measure. However, we have a different risk measure - accumulated costs. We more or less know that when the accumulated costs reach a certain level then we are likely to be unsafe soon. Why not use the costs as distance? Is there a benefit of using this specific risk measure?

---

> ### Author Response · Authors · 2024-11-28
>
> We sincerely thank the reviewer for their thoughtful feedback and for recognizing the value of our work. We are pleased that the motivating example and ablation study were insightful for the reviewer.
>
>
> **Weaknesses:**
>
> **[W1]: Additional Baselines**
>
> We concur with the reviewer’s suggestion to include additional baselines. In response, we have incorporated three baselines: CRPO [1], CVPO [3], and SauteRL [2]. The results for these baselines are presented in Figures 4 and 5 of the revised paper. Additionally, we have expanded our evaluation to include three new environments: SafeMetaDrive, Hopper, and Walker2d. To enhance result visualization, we have included pairwise comparison plots between each baseline and its SRPL counterpart across all environments, provided in Figures 18–21 of the Appendix.
>
> **[W2 & Q1]: Accumulated Cost as a Risk Measure**
>
> Thank you for bringing paper [4] to our attention. We have added it to the Related Work section. Additionally, we have incorporated results for SauteRL [2], which aligns with the concept proposed in [4] and benefits from widely available implementations. SauteRL introduces an accumulated cost-based risk measure by augmenting the state space with this information. However, a notable limitation of this approach is its ineffectiveness when the cost threshold is 1 or lower. In such cases, the accumulated cost remains effectively zero for all states except the one causing a constraint violation, which can hinder effective constraint satisfaction.
> We have included supplementary results demonstrating that while SauteRL performs well on Safety Gym tasks with a cost threshold of 10, it fails to generalize effectively to environments like SafeMetaDrive and AdroitHandPen. These results are presented in Figure 4 of the revised manuscript. Furthermore, we show that integrating SRPL with SauteRL enhances its performance across all evaluated environments.
>
>
> [1] Xu et al. CRPO: A New Approach for Safe Reinforcement Learning with Convergence Guarantee In ICML 2021
>
> [2] Sootla et al. Sauté RL: Almost Surely Safe Reinforcement Learning Using State Augmentation In ICML 2022
>
> [3] Liu et al. Constrained Variational Policy Optimization for Safe Reinforcement Learning In ICML 2022
>
> [4]  Solving Richly Constrained Reinforcement Learning through State Augmentation and Reward Penalties. arXiv preprint arXiv:2301.11592.

---

> > ### Comment · Reviewer_maKe · 2024-11-28
> >
> > Thank you for the changes and the response! I am raising my score
> >
> > PS. I think you are above page limit.

---

### Official Review · Reviewer_K4qc · 2024-11-03

**Soundness:** 3
**Presentation:** 3
**Contribution:** 3
**Rating:** 6
**Confidence:** 3

**Summary:**

This paper introduces the RIPL framework, which integrates state-conditioned risk representations into the agent’s state features. These risk representations help guide exploration more efficiently and reduce the risk of constraint violations without being overly conservative. Empirical results demonstrate that RIPL improves both task performance and safety.

**Strengths:**

1. The idea of state-conditioned risk representations is intuitive and reasonable, and is a novel approach in the community of safety RL.
2. The method is sound and shows significant improvements in multiple environments, and the transferability of the risk models are also tested.
3. The structure of the paper is clear and the writing is easy to follow.

**Weaknesses:**

1. The related works in the recent two years seem to be missing, and the latest baselines that the author compared with are CSC and TRPO-PID, which were published in 2020. Are there other more recent work in the safty RL literature? If time is not enough for experimental comparison, that's fine (though preferred); It would be nice to mention the most recent advances and the advantages of the proposed method compared with them. (I am not an expert in safety RL, but would love to see clarifications on whether there is/is not more recent works)
2. As is discussed in the limitation, the true relation between states' risks might not be genuinely captured (long-horizon dependency etc.). I agree that it is an important aspect to investigate about in future work, but how well can the current risk model reflect the risk? For example, consider providing a visualization of the output distribution of the risk model w.r.t. the constrained states. It'd be nice to have illustrations in the continuous case, similar to Fig.3.

**Questions:**

1. How sensitive does RIPL perform as we set different bin sizes for risk? How do we determine this value?
2. I think transfering the risk model between tasks is an interesting point. Currently, the authors freeze the risk model on the target task. Would it also be possible (and more reasonable) to fine-tune it in the target environment?

---

> ### Author Response · Authors · 2024-11-28
> **Response to Reviewer K4qc**
>
> Thank you for your positive feedback and for acknowledging our contributions. We have revised the paper to address the questions and concerns raised by the reviewer. All modifications to the original manuscript have been highlighted in blue for ease of reference.
>
> **Weaknesses:**
>
> **[W1]: Additional Baselines and Related Work**
>
> Thank you for bringing this to our attention. In response, we have incorporated three additional baselines from recent literature: CRPO [1], SauteRL [2], and CVPO [3]. The results for these baselines are presented in Figures 4 and 5. Furthermore, we have expanded the Related Work section to include more recent research in Safe RL.
>
> **[W2]: Visualization of Safety Representations**
>
> We acknowledge the challenge of modelling long-term dependencies in the current implementation of the safety representation, a difficulty shared by standard reinforcement learning algorithms. We are particularly interested in exploring approaches that can effectively capture long-term dependencies and demonstrate their utility in environments where long-term safety considerations are critical. To illustrate how the risk model—now referred to as the S2C model—assesses the safety of individual states, we have included its output at convergence for three distinct states in the Ant environment. These visualizations are presented in Figure 9 of the Appendix.
>
> **Questions:**
>
> **[Q1]: Sensitivity of RIPL (now SRPL) to Bin Size**
>
> In our experiments, we consistently utilized a bin size of 2 and a safety horizon $H_s$ of 40 steps, except for the Safety Gym environments, where a bin size of 4 and a safety horizon $H_s$ of 80 were employed. We did not perform fine-tuning across different bin sizes or safety horizons. Detailed implementation specifics are provided in Appendix A.1. Additionally, an ablation study exploring various bin sizes and safety horizons is presented in Figures 13 and 14 of the Appendix. The results indicate that smaller bin sizes generally yield superior performance; however, the marginal gains diminish as the bin size decreases further. Figure 14 demonstrates that increasing the safety horizon enhances performance, although the improvement plateaus beyond a certain threshold. This phenomenon can be attributed to the greater significance of short-term safety information over long-term considerations in SRPL agents.
>
> **[Q2]: Finetuning safety representations on the target task**
>
> In response to the reviewer’s feedback, the revised version of the paper now includes results on fine-tuning the risk model—now referred to as the S2C model—on the target task, as presented in Figure 6. The findings demonstrate that fine-tuning the S2C model significantly enhances both sample efficiency and constraint satisfaction. This improvement is particularly notable when transferring both the policy and the S2C model, leading to superior overall performance.
>
> [1] Xu et al. CRPO: A New Approach for Safe Reinforcement Learning with Convergence Guarantee In ICML 2021
>
> [2] Sootla et al. Sauté RL: Almost Surely Safe Reinforcement Learning Using State Augmentation In ICML 2022
>
> [3] Liu et al. Constrained Variational Policy Optimization for Safe Reinforcement Learning In ICML 2022

---

> > ### Author Response · Authors · 2024-12-02
> > **Reviewer K4qc Followup**
> >
> > Dear reviewer K4qc, Thank you once again for your thoughtful review. We have carefully incorporated your suggestions and addressed your questions in the revised manuscript and the comments provided above.
> >
> > As the discussion period draws to a close, we would like to confirm whether you have any remaining concerns. We would be glad to answer any further questions or sincerely appreciate any additional feedback you may have.

---

### Author Response · Authors · 2024-11-28
**General Response to the Reviewers**

We would like to extend our gratitude to the reviewers for their valuable comments and constructive feedback. We are encouraged by their positive reception of our method’s novelty and practicality, as well as their recognition of the paper’s clarity and quality of writing. We have carefully addressed all identified weaknesses and questions, and we provide a broad summary of our revisions below, followed by detailed responses to each reviewer individually. All modifications to the original manuscript have been highlighted in blue for ease of reference.


- We have incorporated several additional baselines, including CRPO [1], SauteRL [2], and CVPO [3], along with new environments such as SafeMetaDrive, Hopper, and Walker2d. Our expanded results demonstrate that our method consistently outperforms baseline approaches across all tested environments.
- We have conducted further ablation studies, including an investigation into the generalizability of the (frozen) safety representations across different cost thresholds, as well as an analysis of the impact of bin size and safety horizon selection for the safety representation.
- We have renamed the method “Safety Representations for Policy Learning (SRPL)” as opposed to the previous “Risk-Informed Policy Learning (RIPL)” and we have renamed the model that learns these safety representations the “steps-to-cost (S2C)” model as opposed to the risk model.  This was done in an attempt to increase clarity and avoid confusion with other established definitions of risk.
- We have included a detailed explanation of the training process for the S2C model to provide further clarity and reproducibility.

Should there be any additional comments, we are available to address them before the end of the discussion period.

[1] Xu et al. CRPO: A New Approach for Safe Reinforcement Learning with Convergence Guarantee In ICML 2021

[2] Sootla et al. Sauté RL: Almost Surely Safe Reinforcement Learning Using State Augmentation In ICML 2022

[3] Liu et al. Constrained Variational Policy Optimization for Safe Reinforcement Learning In ICML 2022

---

### Meta-Review · Area_Chair_MSMW · 2024-12-21

**Metareview:**

Risk Informed Policy Learning for Safer Exploration

Summary: This paper introduces the Safety Representations for Policy Learning (SRPL) framework, addressing challenges in reinforcement learning (RL) for safety-critical environments. Traditional safe RL approaches often fail due to conservative behavior caused by penalties for constraint violations early in training. SRPL tackles this by incorporating state-conditioned safety representations into state features, enabling more balanced exploration and safety. These representations, modeled using a "steps-to-cost" (S2C) approach, estimate the proximity to unsafe states based on past experiences. SRPL enhances the learning process by providing RL agents with improved safety awareness, thus reducing constraint violations and improving task performance. The framework is validated across various tasks, including manipulation, autonomous driving, and navigation, demonstrating improved sample efficiency and safety.

Comments: We received 4 expert reviews, with scores 5, 6, 6, 8, and the average score is 6.25. reviewers have given positive comments about many aspects of this paper. They have mentioned that the state-conditioned risk representations (steps-to-cost) is a novel and intuitive contribution to safe RL, providing agents with better safety awareness without sacrificing task performance. They noted that the SRPL framework is simple to implement and can be integrated with a wide range of RL algorithms, both on-policy and off-policy. The paper is also well-written.

At the same time, there are many weaknesses pointed out by the reviewers. One main weakness is that the paper is missing comparisons with many recent related works such as CVPO and LAMBDA. Similarly, the experiment validation is rather limited, missing benchmarks such as Mujoco and Safe Gym. The reviewers have also asked for additional ablation studies. I acknowledge that the authors have partially addressed these issues during the rebuttal process, but I encourage them to complete this effort while preparing their revised version.

**Additional Comments On Reviewer Discussion:**

Please see the "Comments" in the meta-review.

---

### Decision · Program_Chairs · 2025-01-22

Accept (Poster)